# Oligodendrocyte-mediated myelin plasticity and its role in neural synchronization

Sinisa Pajevic[1]*, Dietmar Plenz[1], Peter J Basser[2], R Douglas Fields[3]

[1]Section on Critical Brain Dynamics, National Institute of Mental Health, NIH, Bethesda, United States; [2]Section on Quantitative Imaging and Tissue Sciences, Eunice Kennedy Shriver National Institute of Child Health and Human Development, NIH, Bethesda, United States; [3]Nervous System Development and Plasticity Section, Eunice Kennedy Shriver National Institute of Child Health and Human Development, NIH, Bethesda, United States

*For correspondence:
pajevic@gmail.com

Competing interest: The authors declare that no competing interests exist.

**Abstract** Temporal synchrony of signals arriving from different neurons or brain regions is essential for proper neural processing. Nevertheless, it is not well understood how such synchrony is achieved and maintained in a complex network of time-delayed neural interactions. Myelin plasticity, accomplished by oligodendrocytes (OLs), has been suggested as an efficient mechanism for controlling timing in brain communications through adaptive changes of axonal conduction velocity and consequently conduction time delays, or latencies; however, local rules and feedback mechanisms that OLs use to achieve synchronization are not known. We propose a mathematical model of oligodendrocyte-mediated myelin plasticity (OMP) in which OLs play an active role in providing such feedback. This is achieved without using arrival times at the synapse or modulatory signaling from astrocytes; instead, it relies on the presence of global and transient OL responses to local action potentials in the axons they myelinate. While inspired by OL morphology, we provide the theoretical underpinnings that motivated the model and explore its performance for a wide range of its parameters. Our results indicate that when the characteristic time of OL's transient intracellular responses to neural spikes is between 10 and 40 ms and the firing rates in individual axons are relatively low (10 Hz), the OMP model efficiently synchronizes correlated and time-locked signals while latencies in axons carrying independent signals are unaffected. This suggests a novel form of selective synchronization in the CNS in which oligodendrocytes play an active role by modulating the conduction delays of correlated spike trains as they traverse to their targets.

## Editor's evaluation

This paper presents a new mathematical model describing biologically plausible feedback that glial cells might use to properly modify the conduction velocity in axons and promote optimal timing of neural impulses through changes in myelination. This work provides an important step forward by providing the theory for myelin-mediated neuronal plasticity.

## Introduction

Temporal precision required in neural processing can range from sub-millisecond in sound localization and echolocation tasks to milliseconds and hundreds of milliseconds in perceptual and motor system signal processing. Often, this is a consequence of individual neural cells or brain regions requiring a narrow time window to integrate signals arriving from multiple sources. Signals traveling from distant

regions commonly traverse complex conduction paths along which conduction velocity (CV) is not constant and undergoes dynamical changes and perturbations, particularly during the development. This will alter the arrival times of action potentials which may undermine the required temporal precision for information processing. It has been argued for more than a decade that a solution to this problem is the adaptive adjustment of the CV through a mechanism of *myelin plasticity* (MP) *Fields, 2005*; *Fields, 2008*, which postulates that myelination is an adaptive and neural activity-dependent process. Modifying myelin sheath thickness and node of Ranvier structure provides the most efficient means to alter conduction delays through changes in CV. There is growing evidence that myelin plasticity is important for fear conditioning *Pan et al., 2020*; *Steadman et al., 2020*, spatial learning *Wang et al., 2020*; *Steadman et al., 2020*, and is shown to be essential for motor skill learning *Bacmeister et al., 2020*; *Kato et al., 2020*; *McKenzie et al., 2014*; *Xiao et al., 2016*. Yet, very little progress has been made in understanding the local learning rules in this new form of plasticity and what feedback oligodendrocytes (OLs) use to properly adjust myelination in the CNS. OLs are mostly located far from the target neurons and lack direct feedback on what the desired CV is because the information about the arrival times of the action potentials, that is spikes, is not available at these intermediate locations. Moreover, in most studies of *activity-dependent* myelination (ADM) *Pajevic et al., 2014*; *Fields, 2015*; *Dutta et al., 2019*; *Stevens et al., 2002*; *Talidou et al., 2021* the precise timing of individual spikes is ignored. It is well known that the introduction of time delays can change both stability and synchronizability at a system level, which then provides an indirect mechanism for MP to affect both the stability *Pajevic et al., 2014* and synchronization, for example in a network of spiking neurons *Talidou et al., 2021*. However, these schemes are based on the activity rate in the connections and do not explicitly include spike timing information in their local rules.

In principle, the arrival times at the target neuron can be explicitly used as the feedback signal, via learning curves similar to that of spike-timing-dependent plasticity (STDP) *Bi and Poo, 1998* but with some important differences. In STDP, the crucial parameter is the pre- and post-synaptic spike time difference, $\Delta t$, the sign of which determines whether long-term potentiation or depression occurs, with $\Delta t = 0$ marking the sharp transition between the two. For MP, such discontinuous learning curves would be unstable and hence must be smoothly ramped across the $\Delta t = 0$ line *Eurich et al., 1999*; *Pajevic et al., 2015*. More importantly, any such feedback information at the target will have to be passed in a retrograde fashion, which is problematic since OLs are mostly located very far from the post-synaptic targets of the axons they myelinate. The same problem applies to schemes in which a network of Kuramoto oscillators is studied and the feedback is based on the phase differences *Noori et al., 2020*.

To develop spike-timing-dependent myelination (STDM) rules, it becomes important to consider schemes in which the mediators of feedback have to act locally and adjust the delays only based on local signal timing information, where the final arrival time error is not available. In this work, we propose models in which OLs are not only the myelinating agents but also serve as the mediators, providing feedback through the interaction with spikes in different axons. We call this form of STDM *oligodendrocyte-mediated myelin plasticity* (OMP). Specifically, we focus on a particular type of OMP, which uses the transient temporal profile of the OL responses to neural spikes as a reference for adjusting CV and relative time delays. We use theoretical arguments, mathematical modeling, and simulations to show that even the simplest form of OMP models can lead to effective synchronization of correlated and time-locked spikes while leaving temporally uncorrelated spikes unaffected.

## Oligodendrocyte-mediated myelin plasticity (OMP)

Our OMP model was inspired by the fact that OL morphology differs drastically from that of Schwann cell (SC), which myelinate axons in the PNS. The most important morphological difference between these two myelinating cell types is that the OL extends many of its processes to myelinate multiple axons, while the SC myelinates only a single axon (*Figure 1A and B*). We hypothesize that this difference comes from the distinct functional roles myelination plays in the PNS and CNS. In the PNS, the goal is to maximize the CV, while in the CNS the presumed goal is to optimize the synchrony of signals arriving from multiple sources. The OL-axon connectivity is schematically depicted in *Figure 1C* and quantified via the *myelination matrix*, $\mathcal{M}$. A single OL can have up to 50 such processes extended to axons within a 100–200 $\mu m$ distance from the soma, but with a tendency to maximize the number of axons it can myelinate, making it unlikely that a given OL would myelinate the same axon twice

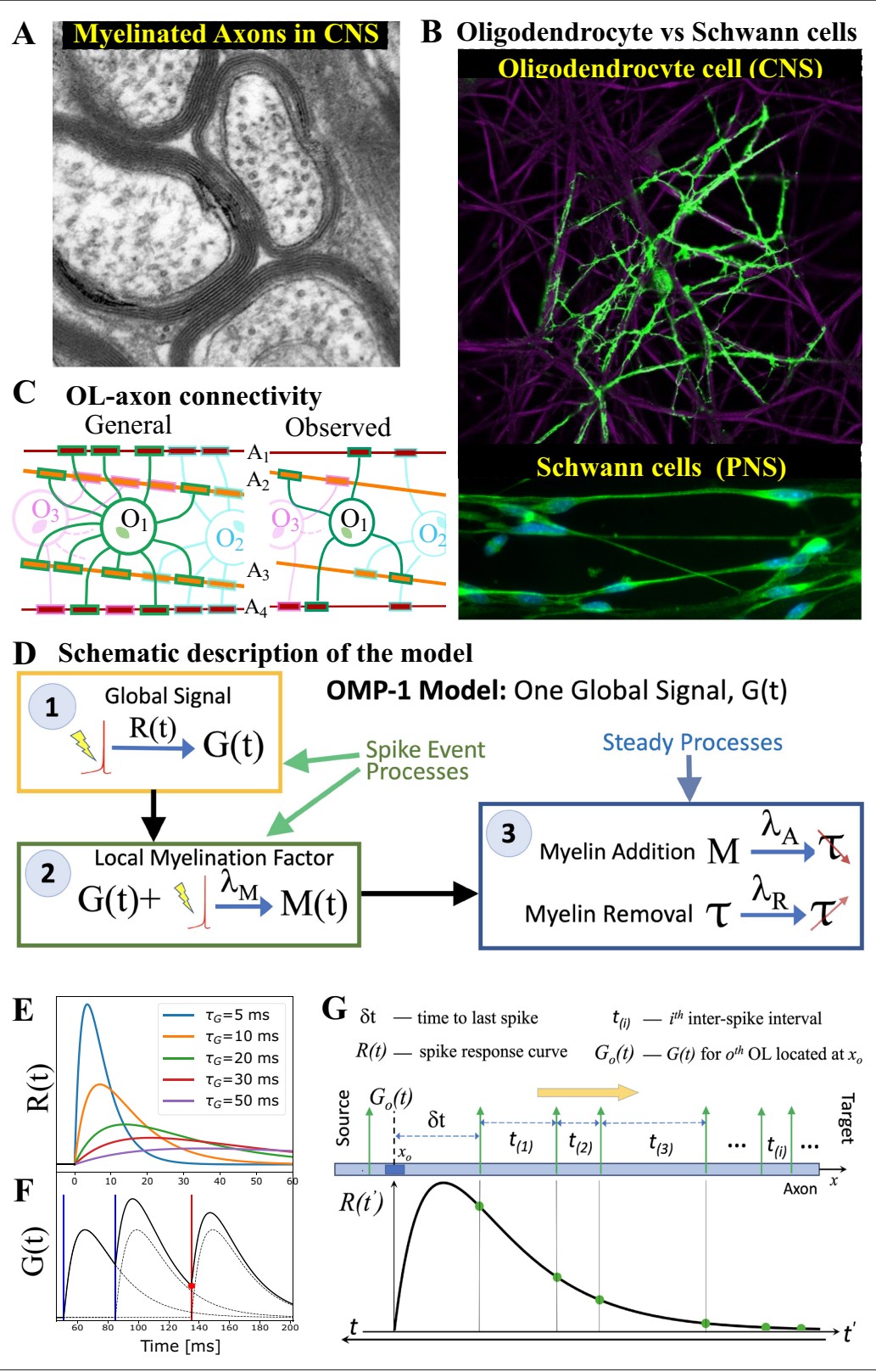

**Figure 1.** OMP motivation and model description. (**A**) Cross-section of myelinated axon bundle in CNS illustrates that myelin thickness on adjacent axons differs. (**B**) Single OL (green) myelinates many different axons (purple), which is in stark contrast to Schwann cells in the PNS, which myelinate only one axon. (**C**) OL-axon connectivity: OLs tend to avoid placing multiple processes on a single axon, thus maximizing the number of axons it myelinates.

*Figure 1 continued on next page*

*Figure 1 continued*

(**D**) Schematic depiction of the OMP-1 model which contains only the three basic steps required for this type of OMP to work. (**E**) Spike-response curves $R(t)$ for increasing values of $\tau_G$. (**F**) The equation governing the release of $G(t)$ is linear and the response to multiple spikes (vertical lines), is the linear sum of individual responses (dashed lines). The release of $M$ after each spike at any given OL process/axon will be proportional to the amplitude of $G(t)$ at the time of spike (red dot for red spike). (**G**) The sum over responses can also be viewed as sampling of $R(t)$ in reverse time. Panels A and B have been adapted from Chapter 45 Figure 1 in *Fields, 2013*, used with permission.

The online version of this article includes the following figure supplement(s) for figure 1:

**Figure supplement 1.** OMP-1 model vs OMP-n model.

**Figure supplement 2.** Time progression of OMP fast and slow variables.

**Figure supplement 3.** Theoretical predictions of OMP model.

**Figure supplement 4.** Theoretical predictions vs simulations.

*Dumas et al., 2015*; *Walsh et al., 2016*. We postulate that this morphology enables a single OL to integrate and compare signals from different axons and act as a mediator in providing the needed feedback for adjusting the relative timing of different signals via dynamic regulation of the CV along these axons. In *Figure 1D*, we outline a general form for such spike-response OMP models, consisting of three essential steps, described below. In these continuous-time models, the transient nature of the temporal profile of the OL response to an action potential plays a crucial role in creating the needed reference for adjusting the relative delays on different axons.

The first essential element of OMP is the release of a *global* intracellular signaling factor, $G$, after each neural spike on any of the axons myelinated by a given OL. This response has a characteristic transient temporal profile, $R(t)$ (Step 1 in *Figure 1D*). When the OL encounters a sequence of spike trains, the resulting global intracellular signal, $G(t)$, will simply be the sum of the individual responses and will fluctuate in time, thus providing a common and time-dependent reference to all of its processes. To allow for differential myelination between different axons, a *local* myelin-promoting factor, $M$, is also required. It is released after each neuronal spike (Step 2) on a given axon $a$, but its release is catalyzed by $G$ and hence is proportional to the global signal, $G(t)$. The OL process that myelinates axon $a$ has a dynamically changing concentration of such local factor, $M_a(t)$, that depends on the temporal profile of the spike trains and will generally differ between the axons myelinated by a given OL. This difference in local concentration of $M$ allows for selective modification of the CV and the conduction delays, $\tau_a$. One can envision more elaborate OMP models (e.g. OMP-n in *Figure 1—figure supplement 1*), in which the release of $M$ can depend on several different global factors, $G_M$, which potentially are released via a cascade of events triggered by the original factor $G$. Here, we use the simplest model of this kind, OMP-1, which has only one global signal, $G$, that does both, responding to neuronal spikes and modulating the local release of $M_a$ at axon $a$, and for the remainder of this work, we are going to refer to the OMP-1 model simply as the OMP model. The last step (Step 3) represents two continuous processes, one being the conversion of $M$ into myelin with some addition rate, $\lambda_A$, and the second being the steady removal of myelin with rate $\lambda_R$ *Dutta et al., 2019*, resulting in time-varying conduction delays, $\tau_a(t)$. The dynamics of the OMP model and its main variables $G(t)$, $M_a(t)$, and $\tau_a(t)$, is governed by a set of equations that implement steps 1–3 (*Equation 8*, *Equation 11*, and *Equation 12*), but also include presumed homeostatic regulation of the myelin conversion/removal rates (*Equation 17*), which is needed for the long-time stability of the model. An example of their time progression is shown in *Figure 1—figure supplement 2*. These equations govern the behavior of a population of OL at a particular segment located at distance $x_o$ along the axonal bundle (see *Figure 1G*). The full OMP model simulates the net delays across a discrete number of such segments, $N_O$, each containing its own set of equations. We call this sequence of OL segments the *oligo-chain* (OC), which is graphically depicted in *Figure 2A*. Based on theoretical arguments elaborated in the next subsection, we expect OMP to synchronize correlated/time-locked signals on different axons while leaving the axons carrying independent signals unaltered (*Figure 2B*), even in situations where individual OL myelinates axons carrying both types of signals (*Figure 2C*). We study its ability to synchronize signals arriving from multiple sources that are temporally dispersed by *fixed delays*, representing persistent relative temporal delays among $N_A$ axons that arise either

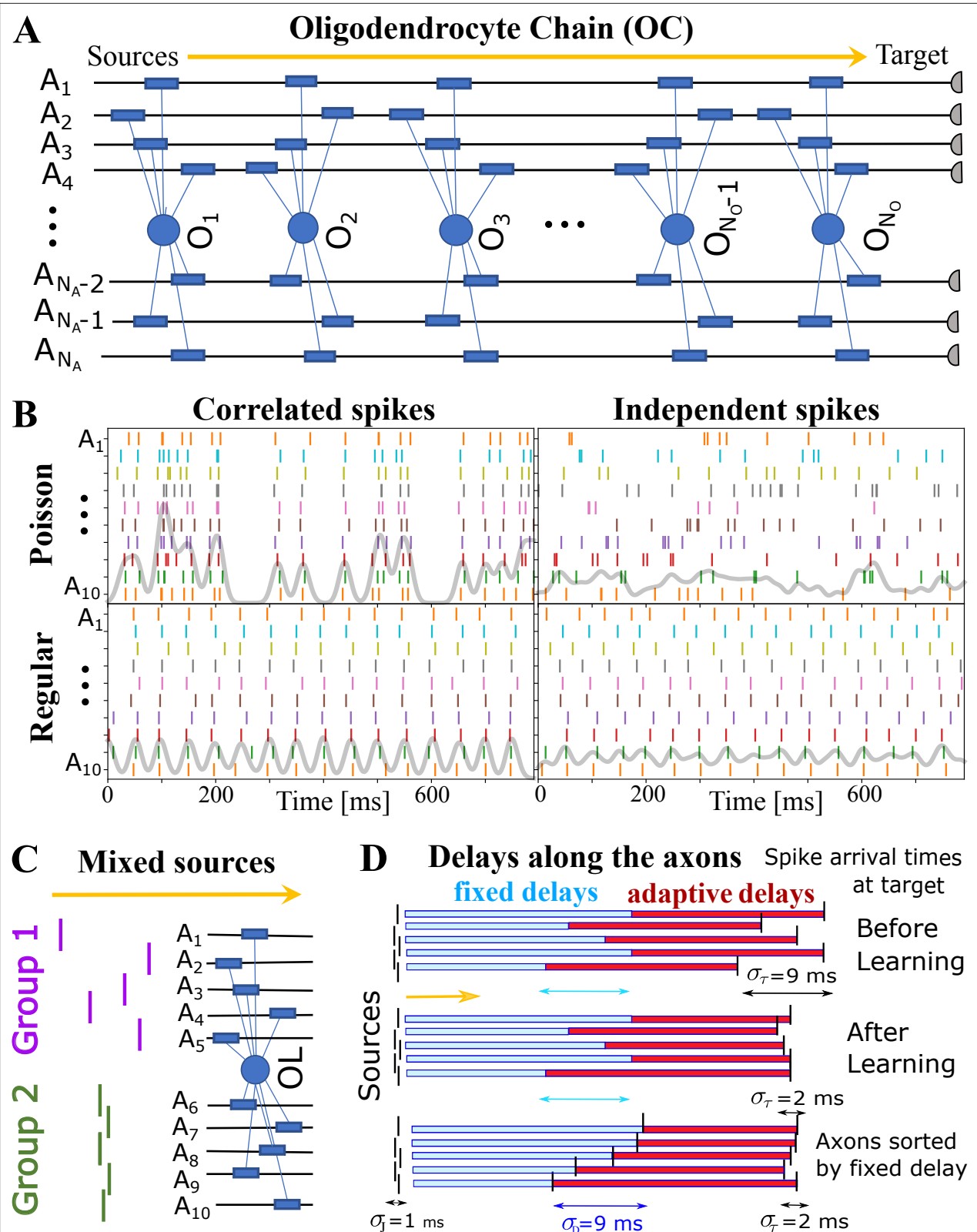

**Figure 2.** OMP simulations with action potentials/spikes conducted along 'oligodendrocyte' chain (OC). (**A**) OC with $N_O$ 'effective oligodendrocytes' (segments), myelinating $N_A$ axons ($\mathcal{M}$ is an $N_O \times N_A$ matrix of ones). (**B**) examples of 'pure' spikes used in our simulations: correlated *versus* independent spikes, and Poisson *versus* regular spiking. Gray lines are moving averages of all spikes using a sliding Gaussian window with 10 ms RMS width. (**C**) example of 'mixed' signals in which axons in Group 1 are conducting independent spikes and those in Group 2 carry correlated spikes. The

*Figure 2 continued on next page*

*Figure 2 continued*

two groups can potentially interfere with the expected synchronization behavior of each 'pure' group. Different groups could also contain spikes time-locked within each group, but independent between the groups. (D) schematic depiction of time delays between the signal source and the target. Fixed delays are not modifiable and essentially represent the spread in spike times as they enter the axonal bundle of myelinated axons, whose delays are adaptive. Note, that the horizontal bars represent the magnitude of the fixed and adaptive delays, not the axons.

The online version of this article includes the following figure supplement(s) for figure 2:

**Figure supplement 1.** Time progression of OMP variables for correlated spikes.

**Figure supplement 2.** Time progression of OMP variables for independent spikes.

from developmental and other structural disturbances in the conduction pathways, or are due to fixed temporal sequences of activations between different sources. We expect those fixed delays, $t_d^{(a)}$, $a = 1, \ldots, N_A$, to be compensated via corresponding *adaptive delays* of the myelinated axonal segments, $\tau_a$, $a = 1, \ldots, N_A$, to facilitate synchronous arrival, as depicted graphically in *Figure 2D*. We test this ability of the OMP model in our simulations that are, together with the details of the model, described in Materials and Methods.

## OMP theory

While OL morphology inspired our OMP model, the motivation was also guided by basic theoretical considerations which provide hints about the model's synchronization performance. We presume that the spike trains in individual axons have the inter-spike intervals (ISIs) that are independent and identically distributed (i.i.d.) with density, $p_{\mathrm{ISI}}(t_{(i)})$, that is, are generated by a renewal process. We denote the mean spiking/firing rate as $f_s$ and the mean inter-spike time interval with $\tau_s = 1/f_s$. The expression for $G_a(t)$, which is a contribution to $G(t)$ coming from spikes on axon $a$, can then be estimated by recognizing that the sum over spikes in *Equation 10* can equivalently be seen as sampling the response function, $R(t)$, in reverse time (see *Figure 1G*). Here, the important parameter is the time to the last spike, $\delta t$, as the remaining spikes are just the sequential samples drawn from $p_{\mathrm{ISI}}(t_{(i)})$, effectively making $G_a(t)$ a function of $\delta t_a = \min|t - t_{k_a}|, \forall t_{k_a} < t$, where $t_{k_a}$ indicates the spike times on axon $a$ for a given OL. If we label the cumulative sum of the subsequent interspike intervals as $t_k^c = \sum_i^k t_{(i)}$, the expression for $G_a(t)$ can be written as

$$G_a(\delta t_a) = R(\delta t_a) + \sum_{k=1}^{\infty} R(\delta t_a + t_k^c). \tag{1}$$

For uncorrelated signals, due to symmetry, any of the axons is equally likely to produce a spike, resulting in equidistributed concentrations of $M$ guided only by the average concentration, $G_{\mathrm{av}} = \langle G(t) \rangle_t$. Predicting the synchronization effects for time-locked signals when the OL myelinates many axons is a more difficult task, particularly when Gaussian *jitter* with spread $\sigma_j$ is added to the specified renewal process ISIs and also due to non-linear saturation effects in the learning equation (see Materials and methods).

When combining $G_a$ from all axons, mutual ordering of spikes will generally need to be considered. In most cases, this derivation will depend on the exact form of $R(t)$ and $p_{\mathrm{ISI}}(t_{(i)})$. For example, for the case of two axons ($N_A = 2$) and a given fixed delay, $t_d$, we need to consider separately the cases where $\delta t \leq t_b = \tau_s - t_d$ and where $\delta t > t_b$ (see *Figure 1—figure supplement 3A*), in order to derive the expressions for $G_{\mathrm{av}}$, $\langle M_1 \rangle$, $\langle M_2 \rangle$, as a function of $\tau_s$, $\tau_G$, and $t_d$. For $\tau_s < 2t_d$, the "leading edge" axon becomes the follower, in which case the myelination pattern is reversed, making its CV faster instead of slower (*Figure 1—figure supplement 3B, C*). We demonstrate such calculation in the Appendix B for a simple case with $N_A = 2$, regular spiking, and without jitter. The calculations become more cumbersome with increasing $N_A$, even when jitter is ignored, and this is true for most of the renewal processes governing the spiking dynamics on a given axon. However, in the case of a pure Poisson process ($p_{\mathrm{ISI}}(t) = e^{-t/\tau_s}$), we can utilize its memoryless property to derive simple expressions for $G_{\mathrm{av}}$ and $\langle M_a \rangle$. The expression for $G_{\mathrm{av}}$, at any of the locations along the axons, is simply

$$G_{\mathrm{av}} = \langle G(t) \rangle_t = N_A/\tau_s. \tag{2}$$

The expectation for $M$ in the case of Poisson spiking is also simple. For example, when $N_A = 2$, the average concentration of $M$ at the leading edge axon is $\langle M_1(t) \rangle = C_M G_{av}$ while for the lagging axon $\langle M_2(t) \rangle = C_M(G_{av} + R(t_d))$, where $C_M = \lambda_M/(\lambda_A \tau_s)$. Hence, the ratio of their concentrations is always lower than one,

$$r_2 = \frac{M_1}{M_2} = \frac{G_{av}}{G_{av} + R(t_d)} = \frac{2}{2 + \tau_s R(t_d)} < 1, \tag{3}$$

which will consistently lead to a greater increase in CV of the lagging axon, hence supporting synchronization. This also reveals the landscape of OMP synchronization, expected to be most efficient for $r_2 \ll 1$, which happens at low firing rates (large $\tau_s$) and a fast spike response time, $\tau_G$ (*Figure 1—figure supplement 3C*). In the case of multiple axons, the above argument retains its simplicity and we can write the expected time-averaged concentration of $M$ on axon $a$ as

$$\langle M_a \rangle_t = C_M \left( G_{av} + \sum_{t_d^{(i)} > t_d^{(a)}} R(t_d^{(i)} - t_d^{(a)}) \right). \tag{4}$$

The *Equation 4* is an important result that indicates that with Poisson spiking and fixed time delays along different axons, the differential expression of the myelination factor will always myelinate the lagging axons more than the preceding ones. The expression in *Equation 4* assumes no jitter but still agrees well with the simulated values for $\sigma_j < 3$ ms (*Figure 1—figure supplement 4*).

## Results

Our measure for quantifying the synchronization properties of the OMP model is the spread in the spike arrival times across all axons, $\sigma_\tau = \mathrm{SD}_a(D_a + \tau_a)$, where $D_a$ stands for pre-specified fixed delays (normalized to some prescribed value, $\sigma_D$), and $\tau_a$ are the adaptive delays (see *Figure 2D*). The spread starts with some large value, mostly due to the spread among fixed delays, $\sigma_\tau^{(0)} \approx \sigma_D = \mathrm{SD}_a(D_a)$, and in the course of time, due to the OMP dynamics, is reduced to lower values, ideally to zero, indicating perfect synchronization. We collect the time course of $\sigma_\tau$ during learning, and we call it a synchronization profile, $\sigma_\tau(t)$. We study the performance of the OMP model by characterizing the synchronization profiles, $\sigma_\tau(t)$, obtained for a wide range of the OMP model parameter values, utilizing a grid-search-like exploration. The sets of parameter values explored in these simulations are provided in Appendix 1, while the summary description of the OMP model parameters is given in *Table 1*. In most studies, we obtained a large number of synchronization profiles, $\sigma_\tau(t)$, one for each parameter setting/set, which were then reported as averages over all runs, but grouped by a given parameter of interest, or characterized using a model fitting and selection procedure described in Materials and methods (see also *Figure 3—figure supplement 1C*). In particular, we focus on the long-time baseline parameter, $\sigma_\tau^\infty = \lim_{t \to \infty} \sigma_\tau(t)$ which represents the long-time ability of the OMP model to synchronize signals for a given parameter setting.

We show in *Figure 3A* the estimated distribution of $\sigma_\tau^\infty$, as well as the model selection chart, when fitting a large number ($n = 17280$) of synchronization profiles, $\sigma_\tau(t)$, obtained using a wide range of OMP parameters (*Appendix 1—table 1*, *Figure 3—figure supplement 2*, and *Figure 3—figure supplement 1* for details). The results indicate that the OMP model behaves as desired. When the myelinated axons conducted correlated spikes, we observed a significant reduction in the conduction delay spread, $\sigma_\tau$, that is, a significant increase in synchronization. No change was observed for independent spikes resulting in synchronization profiles best fit to the constant model, C (purple) (see Model Fitting for synchronization profiles, *Figure 3—figure supplement 2*). For correlated spikes, a single exponential approach to synchrony was most commonly observed (E1). In several instances, $\sigma_\tau(t)$, was not monotonic and sometimes appeared oscillatory, which can be seen upon inspecting individual runs, particularly for small $N_O$ ($N_O < 3$), short $\tau_G$, and small jitter, $\sigma_j$ (*Figure 3B*). As suspected, the OMP model with only a single OL has an inherent instability due to the presence of fixed delays, which could be alleviated by increasing $\tau_G$ and $\sigma_j$. This instability, however, rapidly disappeared for any level of jitter when longer OCs are used (*Figure 3B*, solid lines; see also *Figure 1—figure supplement 1* and *Figure 4—figure supplement 3*). These trends can be seen in *Figure 3C*, where the averages of all $\sigma_\tau(t)$ profiles grouped by $N_O$ are shown. The runs for small $N_O$ were longer, to match

**Table 1.** List of symbols used in the manuscript: OMP model parameters, OMP variables, spiking signal parameters, and data analysis and quantification parameters and other symbols.

For each we provide a short description and the range of the parameter values, or its dimensionality, in the case of the variables.

**List of symbols used in this manuscript**

| Symbol | Description | Explored Values/Dimensionality |
|---|---|---|
| **OMP model parameters** | | |
| $R(t)$ | OL transient response curve | see $\tau_G$ |
| $\tau_G$ | characteristic time for $R(t)$ ($\tau_G = \tau_r = \tau_d$) | [2-100] ms |
| $\lambda_M$ | production rate for factor $M$ | [0.01, 0.5] ms$^{-1}$ |
| $\lambda_A$ | myelin conversion/addition rate | [0.01, 0.5] ms$^{-1}$ |
| $\lambda_R$ | myelin removal rate | **Equation 16**; variable for $\lambda_H > 0$ |
| $\lambda_H$ | homeostatic rate | [0, 10$^{-2}$] ms$^{-2}$ |
| $N_O$ | number of OL in OC | [1-20] |
| $N_A$ | number of axons | [2-150] |
| $\mathcal{M}$ | myelination matrix | $N_O \times N_A$: {full connectivity} |
| $\tau_{\min}$ | minimal delay attainable on axons | 3 ms |
| $\tau_{\max}$ | maximal delay attainable on axons | 100 ms |
| $\tau_{\text{nom}}$ | nominal/homeostatic delay | [10-90] ms |
| $\tau_0$ | initial adaptive delays parameter | $\tau_0 = \tau_{\text{nom}}$ |
| $p_\tau$ | percent spread of initial adaptive delays | 5% |
| **OMP model variables** | | |
| $G(t)$ | global OL signal (**Equation 8**) | $N_O$ variables |
| $M_a(t)$ | concentration of $M$ on axon $a$ (**Equation 11**) | $N_O \times N_A$ variables |
| $\tau_a(t)$ | conduction delay(s) for axon $a$ (**Equation 12**) | $N_A$ or $N_O \times N_A$ variables |
| $\tau_a^o(t)$ | $\tau_a$ for OC segment $o$ (**Equation 12**) | $N_O \times N_A$ variables |
| $\lambda_R(t)$ | myelin removal rate (**Equation 17**) | OMP parameter when $\lambda_H = 0$ |
| **Spiking signal parameters** | | |
| $\tau_s$ | inter-spike time interval parameter | [10-250] ms |
| $f_s$ | mean firing rate | [4-100] Hz |
| $t_R$ | refractory period in refractory Poisson process | [0–100] ms |
| $\sigma_j$ | amount of jitter given to spikes | [0–10] ms |
| $\sigma_D$ | SD for fixed delays | [0–20 ] ms |
| $\sigma_s$ | percent variability in firing rates | [0–20] % |
| $T_{\exp}$ | total duration of simulation | [10 min - 10 hrs ] |
| $n_e$ | number of recorded epochs during simulation | [20–500 ] |
| $T_e$ | duration of each recorded epochs | $T_{\exp}/n_e$ [10 sec - 5 min] |
| $n_r$ | number of replicated simulations/trials | [3 - 10] |
| **Quantification parameters/other symbols** | | |

*Table 1 continued on next page*

*Table 1 continued*

**List of symbols used in this manuscript**

| | | |
|---|---|---|
| $\sigma_\tau$ | OC spread SD ($D_a + \tau_a$) | evaluated after each epoch |
| $\sigma_\tau(t)$ | synchronization profile during OMP learning | $\sigma_\tau$ values for $n_e$ epochs |
| $\sigma_\tau^{(0)}$ | initial spread before learning, $\sigma_\tau^{(0)} = \sigma_\tau(0)$ | determined by $\sigma_D$, $\tau_0$, and $p_\tau$ |
| $\sigma_\tau^\infty$ | long-time baseline, $\sigma_\tau^\infty = \lim_{t\to\infty} \sigma_\tau(t)$ | estimated via model fitting |
| $\tau_L$ | characteristic time for synchronization | estimated via model fitting |
| $L_\tau$ | learning/synchronization rate | $= 1/\tau_L$ |
| $t_d$ | generic name for fixed delays | NA |
| $D_a$ | pre-specified/normalized fixed delay on axon $a$ | random values, $\mathcal{N}(0, \sigma_D)$ |
| $t_{(i)}$ | ISI for the $i^{th}$ interval | NA |
| $p_{\text{ISI}}(t_{(i)})$ | probability density function of ISI | NA |

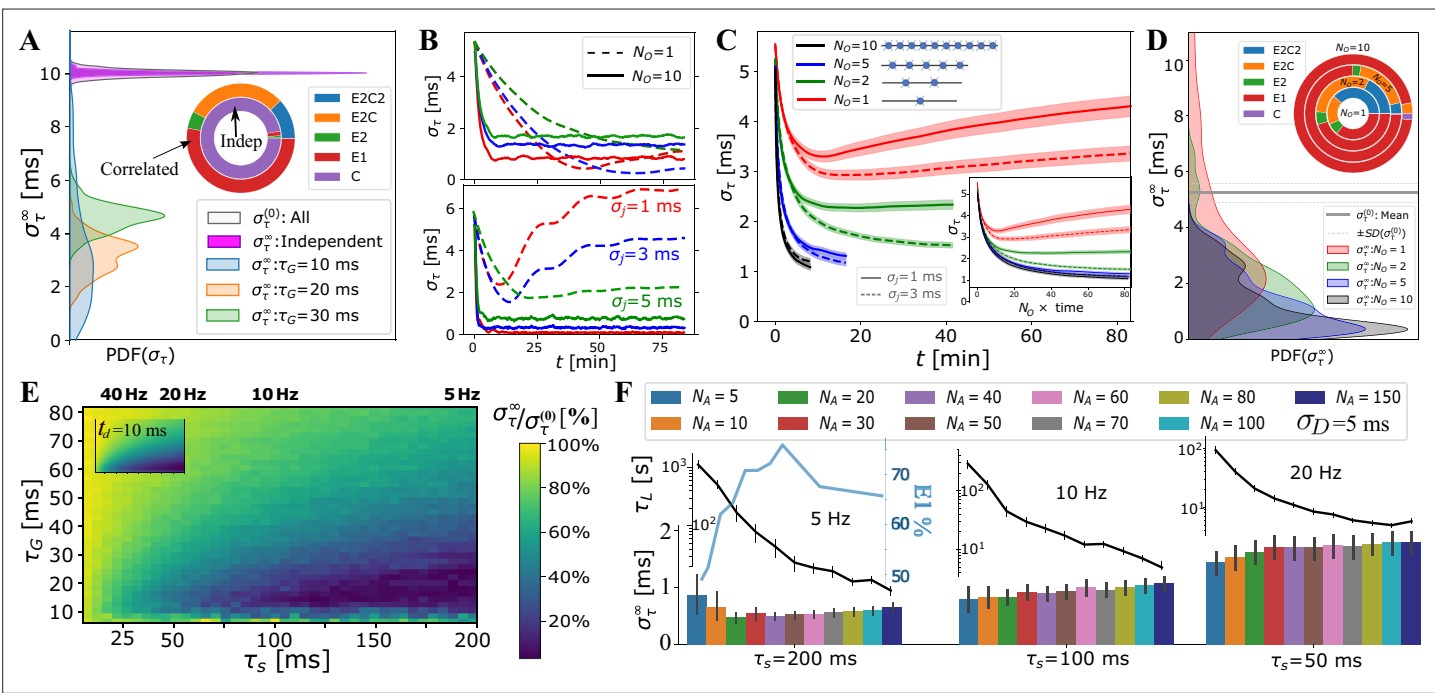

**Figure 3.** OMP model behavior as a function of OMP and signal parameters. (**A**) When spikes are independent, $\sigma_\tau(t)$ consistently show no change in overall synchronization (purple; innermost circle in the model selection pie-chart); for correlated signals significant synchronization occurs, largely depending on $\tau_s$ and $\tau_G$. (**B**) individual OMP synchronization profiles, $\sigma_\tau(t)$, simulations for $N_A = 10$, $\lambda_H = 10^{-6}$, $\lambda_M = 0.02$, with $\tau_G = 30$ ms (top panel) vs. $\tau_G = 10$ ms (bottom). Dashed lines are for $N_O = 1$ and solid lines for $N_O = 10$. Colors indicate the jitter level: $\sigma_j = 1$ ms (red), $\sigma_j = 3$ ms (blue), $\sigma_j = 5$ ms (green). (**C**) dependence of $\sigma_\tau(t)$ on the number of OLs, $N_O$, in the OC. We show both the comparison based on the raw time, as well as when matched in terms of the total number of neural spikes encountered by the OLs (inset). (**D**) density of $\sigma_\tau^\infty$ estimates dependence of $\sigma_\tau$ on the number of OL in OC. (**E**) Percent reduction in $\sigma_\tau$ vs $\tau_G$ and $\tau_s$ for $\sigma_D = 10$ ms; (inset) ratio of the $M$ concentrations for two axon case, $N_A = 2$, which matches well the overall pattern of synchronization. (**F**) dependence of $\sigma_\tau^\infty$ (bar plot) and the synchronization time constant, $\tau_L = 1/L_\tau$, (semi-log plot, above) on the number of axons, $N_A$ values are only for runs declared as E1 blue curve inset indicates percent of E1, for different $N_A$ (for all spike rates).

The online version of this article includes the following figure supplement(s) for figure 3:

**Figure supplement 1.** Exploring OMP properties and parameters.

**Figure supplement 2.** Model fitting for synchronization profiles.

them in terms of the number of spikes processed. When plotted in actual time, it becomes evident that having more OLs in an OC greatly increases OC stability as well as the synchronization/learning rate, $L_\tau$. Since the averages were obtained over a large number of runs, the standard error (SE) is small and the oscillatory or other forms of instabilities average out. To better quantify all $\sigma_\tau(t)$ s we fit them to five different models, as described in Methods. In **Figure 3D**, we show the $\sigma_\tau^\infty$ distributions and the model selection chart for different values of $N_O$. They all indicate that, with increasing $N_O$, synchrony is improved and instabilities disappear. For $N_O = 10$ most of the OMP simulations yielded a stable exponential decay to a synchronized state (73% of all runs reduced the arrival time spread from the initial $\sigma_\tau^{(0)} = 10$ ms to below 3 ms; for $\tau_G = 10$ ms this increases to 97%, with 74% synchronized below 1 ms).

For correlated spikes, the effectiveness of synchronization strongly depended on two temporal parameters: the characteristic response time of the OL, $\tau_G$, and the mean inter-spike interval, $\tau_s$, which was assumed to be the same for all axons. In **Figure 3E**, we explored the synchronization effect as a function of $\tau_G$ and $\tau_s$; the results shown are for $N_O = 5$, $\sigma_j = 1$ (see **Appendix 1—table 1**). For very short $\tau_G < 10$ ms, performance was unstable, reflecting the fact that for the short-lasting spike responses, $R(t)$, the comparison window between spikes in different axons is too narrow. On the other hand, having spike responses last too long, i.e., $\tau_G > 40$ ms, makes the resulting $G(t)$ too smooth to differentially release $M$, particularly for small $\tau_s$ (**Figure 1—figure supplement 3B, C**). Accordingly, we found that for intermediate values of $\tau_G \in [10 - 40]$ ms, synchronization was predictably achieved and was highly efficient for firing rates $f_s < 10$ Hz.

In **Figure 3F**, we study the dependence of $\sigma_\tau^\infty$ and the synchronization time constant, $\tau_L = 1/L_\tau$, on the number of axons that OLs myelinate, $N_A$. The results are grouped by the firing rate simulated. For a low firing rate of 5 Hz, optimal performance was achieved for $N_A = 20$, whereas synchronization became progressively harder with increase in firing rate and number of axons to synchronize. In general, increasing $N_A$ sped up synchronization for all firing rates explored. The fastest synchronization achieved at 5 Hz for an intermediate number of axons ($30 < N_A < 60$) also coincided with the highest fraction of stable, E1, synchronization profiles (peaking at $N_A = 50$, light-blue). We note that myelinating more than $N_A = 50$ axons did not improve synchronization efficiency for any of the spiking rates, which, perhaps, hints to why OLs rarely extend more than 50 processes.

In **Figure 4A-C**, we tested the ability of the OMP model to selectively handle mixed sources of signals by having non-overlapping groups of axons carry spikes with different $p_{\text{ISI}}$, or different mutual correlations. Spikes from different groups are inducing responses in the same OL, and hence mutually interfere and can potentially corrupt the expected behavior for the equivalent 'pure' group, that is, disrupt OMP's ability to synchronize the correlated group of signals, or erroneously synchronize the independent signals. In **Figure 4A and B**, we show the results when one group of axons carries correlated and another carries independent spikes. In **Figure 4C**, we explore the effect of different groups carrying signals that are correlated within each group, but not between groups. We evaluate $\sigma_\tau$ within each group and compare it with the equivalent pure signal sources, the number of axons being matched in all comparisons. We found an increase in synchronization only for correlated groups, with only slight 'jamming' of the synchronization performance due to the presence of other groups with potentially corrupting signals. The independent signals remain unaffected, which is a desired behavior for the selective synchronization of spikes from different neuronal populations.

For OMP to be operational requires an overall balance between myelin removal (controlled by $\lambda_R$) and myelin addition (controlled by $\lambda_M$ and $\lambda_A$). For example, if $\lambda_R$ is too large the long-term behavior of the OMP model would lead to complete myelin removal. Here, we achieve such an operational regime by treating $\lambda_R$ as a variable and control its rate of change with the homeostatic rate, $\lambda_H$, as defined in the homeostatic equation (**Equation 17**). The results in **Figure 4—figure supplement 2C** indicate that this homeostatic regulation, while keeping the model operational, does not play an essential role in OMP synchronization. Although it appears that for low firing rate and for large fixed delays ($\sigma_D = 10$ ms), increasing the value of $\lambda_H$ has some detrimental effect on model performance, this effect is not seen for $\sigma_D = 5$ ms. The oscillatory dynamics of $\lambda_R$ depends on $\lambda_H$ and $\lambda_M$ (see **Figure 4—figure supplement 3**) which can propagate into synchronization profiles. Nevertheless, the pie-chart in **Figure 4E** does not indicate that $\lambda_H$ strongly influences the model selection, independent of what value of $p_{\text{MSE}}$ was used. **Figure 2—figure supplement 1**, **Figure 4—figure supplement 3** show that the oscillations roughly average to the same value and thus do not significantly affect synchronizability in the long term. We note that the average values of $\lambda_R$ deviate from the expected

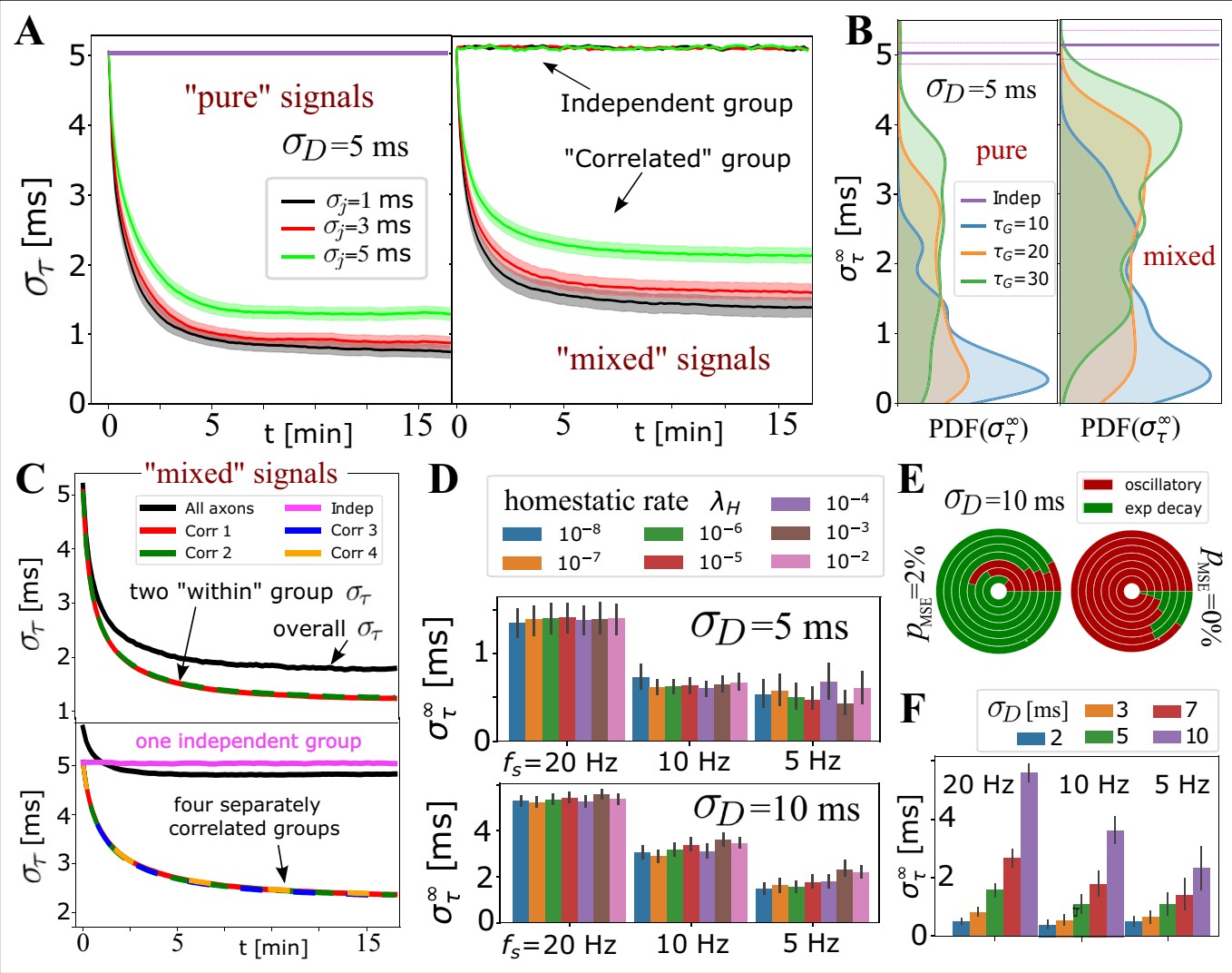

**Figure 4.** Selective synchronization with mixed signals; effects of $\lambda_H$ and $\sigma_D$ parameters. (**A**) Average $\sigma_\tau(t)$ (grouped by $\sigma_j$; shaded regions indicate SE over the parameter sets for $N_A = 10$ and $\sigma_D = 5$; see **Appendix 1—table 2**). In the left panel is the average for the set of $N_A = 10$ axons conducting only 'pure' correlated signals, and on the right are the averages within two groups of 10 axons, out of $N_A = 20$ total that OL myelinates, carrying 'mixed' signals – one group conducting correlated signals and the other independent signals. The behavior of the correlated groups is clearly distinguishable from the independent ones (indicated by arrows). (**B**) Estimated density for $\sigma_\tau^\infty$ comparing pure (left) vs mixed signals (right panel) for a wider range of parameters, including $N_A = 20$ and $N_A = 50$, but with the number of axons carrying correlated signals matched in all cases ($N_A = 10$, $N_A = 25$). (**C**) Top panel shows $\sigma_\tau(t)$ for two groups of correlated signals (colored lines) with Poisson spiking (correlated within group, but mutually independent); bottom panel shows results for four separately correlated groups + 1 independent group (cyan). Black lines represent the spread over all axons. (**D**) Investigating the influence of homeostatic regulation and homeostatic rate, $\lambda_H$, grouped by different firing rates and different $\sigma_D$. (**E**) Model selection chart depicting the relative proportions of oscillatory (E2C2, E2C) vs stable, single exponential (E1) $\sigma_\tau(t)$, for two different values of the F-test fudge-factor $p_{MSE}$; the innermost circle is for $\lambda_H = 10^{-2}$ and the outermost for $\lambda_H = 10^{-8}$. (**F**) OMP synchronization performance for different magnitudes of fixed delays, $\sigma_D$.

The online version of this article includes the following figure supplement(s) for figure 4:

**Figure supplement 1.** Supplemental results for **Figure 4**.

**Figure supplement 2.** Influence of additional OMP and signal parameters.

**Figure supplement 3.** Oscillations due to homeostatic regulation.

value given by **Equation 16**, even for the case of independent spikes (**Figure 2—figure supplement 2**). This difference arises from ignoring saturation effects and depends on the values of $\tau_{nom}$, $\tau_{min}$, and $\tau_{max}$. We also explored scenarios in which we use the 'true' balancing homeostatic value of $\lambda_R$, obtained via a trial run, and set $\lambda_H = 0$. The corresponding synchronization profiles were not

significantly affected, further indicating that the homeostatic process is not an essential element for achieving spike synchronization in the OMP model.

The production rate of $M$, $\lambda_M$, while greatly influencing the learning rate, $L_\tau$, has negligible influence on $\sigma_\tau^\infty$ (*Figure 4—figure supplement 2A and B*). Similarly, the conversion rate $\lambda_A$ had very little influence on the outcome of OMP (*Figure 4—figure supplement 2D*), indicating that the simplified OMP model with instantaneous myelination could be a more efficient way of studying its behavior (*Figure 3—figure supplement 1E*). In *Figure 4F*, we explored the dependence of $\sigma_\tau^\infty$ on the magnitude of fixed delays, $\sigma_D$. For correlated signals, synchronization always occurs but its efficiency decreases in terms of $\sigma_\tau^\infty/\sigma_\tau^{(0)}$ when $\sigma_D$ becomes large (for $\sigma_D > 10$ ms see *Figure 4—figure supplement 1B*). Such large delays might be commensurate with the delay corrections needed during the development but are presumably much larger than the timing corrections needed in the adult brain.

## Discussion

Here, we report the development of a simple, biologically plausible model of oligodendrocyte-mediated myelin plasticity that synchronizes temporally correlated neuronal spikes, as they travel along an axonal bundle, while leaving temporally independent spikes unaffected. This enables the OLs in the model to counteract the temporal dispersion arising from heterogeneous conduction delays and facilitate synchronous arrival times of spikes coming from distant neuronal populations with correlated activity. The general idea of myelin plasticity is not new *Fields, 2005*; *Fields, 2008*, however, a local STDM mechanism by which the brain could robustly and selectively adjust axonal latencies has been missing. Our OMP model introduces robust local learning rules and feedback mechanisms for adaptive changes that yield desired results under a wide range of biologically realistic parameters. The adaptive changes to axonal delays are selectively applied to groups of axons that carry correlated spikes so that they arrive at their targets simultaneously, while those carrying independent or uncorrelated spikes are not affected (*Figure 4A–C*). This selectivity conforms to known relationship between circuit anatomy and function, in which neighboring axons share similar temporal firing and functional properties; for example, the tonotopic organization of auditory cortex and the cortical homunculus in somatosensory cortex. Such correlated firing also drives refinement of connections between the retina and LGN during development *Meister et al., 1991*. Myelination usually begins after axons reach their target and become functional, starting from the cell body and proceeding toward the axon terminal. This is clearly evident in the optic nerve, where OL progenitor cells migrate out of the brain and into the optic nerve during development, yet axonal myelination proceeds in the reverse direction, beginning at the retina on retinal ganglion cells and proceeding toward the optic chiasm *Ishibashi et al., 2009*. Such a proximal-to-distal gradient agrees well with our OMP model as the OLs at the source end of the OC will experience less synchronized signals. Previous work that use phase- and time-dependent models of myelin plasticity *Noori et al., 2020*; *Pajevic et al., 2015* presume a priori that the temporal difference feedback at the target is available to OLs, however, this local information at the target, that is, synapse, will have to be transported in a retrograde fashion. Besides being slow, such a process would suggest that more myelin will be found close to the target rather than far away from it. We note, that with the existence of targeted fast axonal transport, for example via mitochondria, it is still possible that any myelin-promoting factor is transported in a retrograde fashion far from the synapse.

The fact that OLs with more than 50 processes are uncommon is also in accord with our observation that having $N_A > 50$ is not advantageous, according to our OMP results. Another testable prediction of this model would be that structural deformation or lesion to a portion of axons carrying correlated activity would lead to re-myelination downstream or immediately after the site of the lesion, while myelin upstream will not be affected. Results in *Figure 4* suggest that the efficiency of synchronization is slightly reduced when a fraction of axons carry uncorrelated signals. Hence, we postulate that in the brain, OLs might be removing their processes from such axons while keeping or growing new processes only on those axons that carry synchronized signals. OMP predicts that synchronization is most effective at lower firing rates, and thus we expect it to occur when the brain operates in low spiking rate regimes.

It was observed that OLs can undergo sudden depolarization which leads to significant changes (10%) in *Yamazaki et al., 2007*, which might suggest that such events are essential for efficient MP. While this can be incorporated into a general OMP scheme (*Figure 1D*), the OMP-1 shows that

passive responses of OL cells are sufficient for synchronizing neural signals, as well that astrocytes, commonly considered the actors in providing the feedback in MP, are not needed. Our results also demonstrate that the presence of noise acts as a stabilizer, both in terms of jitter removing the instabilities in the OMP model as well as Poisson spike dynamics always biasing larger concentrations of the $M$ to be on the axons with larger fixed delays, as opposed to regular spiking which can reverse this pattern and destabilize the system. Hence, Poisson spiking, even though slower and in many situations less efficient than regular spiking, is more reliable for the MP. Our model of synchronization provides further support for the narrative that the noisy brain is a healthy brain, and too much regularity/synchronization in the brain dynamics can lead to its failure.

The OMP model described here, while simple, is not the simplest form that can serve as a proof of concept. An instantaneous myelination model described in Methods, also shows robust synchronization performance (*Figure 3—figure supplement 1E*), with shortcomings coming only at high learning rates (large $\lambda_M$) and when using homeostatic regulation in *Equation 17*, making it then a stochastic equation. But, we kept a more general form, since we expect that in future developments a more sophisticated regulation of $\lambda_M$, $\lambda_A$, and $\lambda_R$ will be needed, to account for elaborate time-locking patterns where the firing frequencies are different but matched via integer multiples. One of the weaknesses of the current model is that $\lambda_M$ is constant and the same for all axons, making the model sensitive to rate differences, which can override the synchronization effects. To make it work, the production rates, $\lambda_M$, need to have their own homeostatic regulation, so that the axons with higher firing rate will down-regulate their $\lambda_M$. In more elaborate OMP models a mix of activity- and time-dependent MP might be needed. The net result will depend on the relative strengths/learning rates between the time-dependent and the activity-dependent learning, something that will require its own independent study. For the current model, we explored its sensitivity to firing rate variability (*Figure 4—figure supplement 2E and F*) which showed that variations in firing rate greater than 5% are highly detrimental to OMP's ability to adjust fixed delays properly. When the firing rates across axons are very different, it might also not be desirable to synchronize those groups of axons, nor is easy to define the temporal synchrony in such a situation.

There are other aspects of OMP that are not explored here. For example, the effects of inhomogeneous OL-axon connectivity are not addressed, as we use the same myelination matrix along the OC. The stochasticity in our models comes mainly from the stochasticity of the spikes and their jitter. Future work will address complex patterns of connectivity, other independent sources of noise for both global and local factors, as well as more sophisticated homeostatic regulation discussed above. Delays in factor $G$ are mainly implemented here through the rise time, $\tau_r = \tau_G$, however, an increase in $G$ coming from a particular OL process will not affect all processes simultaneously, and the relative delays between different processes can have significant effects on the resulting synchronization, which will need to be explored via delay-differential equations, or using a discrete implementation. Some of the proposed mechanisms for myelin plasticity are discrete in nature, for example, the treadmilling model *Dutta et al., 2019*, but the discreteness is only in the state of myelination; the feedback mechanisms needed for temporal adjustments in any time-dependent MP might still need the continuous-time comparisons. Discretizing time in the simulations can introduce its own effects, which could dominate synchronization performance since MP mechanisms usually require long-time simulations (the MP time scale is many orders of magnitude longer than the neuronal spiking time scale).

In summary, we demonstrated that the simple and biologically plausible adaptive dynamics of the OMP model leads to efficient and selective synchronization of correlated and time-locked signals, without affecting mutually independent streams. This is a novel perspective in brain organization in which white matter conduction properties and myelin plasticity act as a temporal 'lens' to 'focus' multiple spike trains as they target a particular brain region. Our model also addresses lingering questions about the illusive local learning rules and feedback mechanisms in MP, as it circumvents the need for direct information about the actual arrival times at the target. With its precisely spelled-out dynamics and learning rules, it serves as a useful basis for designing future experimental tests aiming to elucidate the nature of MP in the CNS.

## Ideas and speculation

The main goal of our OMP model is to conceptualize a general but biologically plausible myelin plasticity mechanism by which synchronization can be achieved. In the Discussion, we made suggestions

about some testable predictions of our model and here we additionally speculate on the biological aspects of our mathematical model, in particular, we discuss the potential candidates for the factor $M$ and the global signal $G$. Our usage of the terms somewhat implies that $M$ is a molecular factor, while $G$ is some fast propagating signal, for example, intracellular potential or ionic concentration, which is mainly based on the timing constraints implied by our model. It suggests that the release and clearance of a global intracellular signal cannot be too slow since, according to the OMP model, synchronization is not very efficient when the characteristic time for the release and clearance of $G$, $\tau_G$, is larger than 80 ms, even for slow firing rates. But it also suggests that the clearance does not have to be too rapid; in fact, having $\tau_G$ too short can be detrimental in some situations. For firing rates below 10 Hz, $\tau_G$ should ideally be in the range 10 ms $< \tau_G <$ 40 ms for effective synchronization of correlated inputs even for large fixed delays (e.g., $\sigma_D =$ 10 ms). These timing requirements make intracellular Ca2+ a good candidate for the role of $G$, since it is also a catalyst for many bio-molecular reactions. While at present we only speculate that this is the case, we also emphasize here some of the established biological evidence regarding signaling mechanisms between axons and OL, as well as highlight some of the difficulties in conducting experimental tests of our model.

OLs express many of the same neurotransmitter receptors and ion channels that are expressed by neurons, enabling robust activity-dependent axon-OL communication through several signaling mechanisms that differ depending on the developmental stage of the cells. Several types of neurotransmitter receptors have been identified on the axon underlying compact myelin *Stys, 2011*; however, the detection of local signaling events between axons and mature OLs with compact myelin is difficult with current methods. Depolarization at the distal tips of the OL processes that are in contact with axons is not accessible for measurement by patch electrode recording at the cell body because of the electrotonic decay over the long slender cell process. Calcium imaging using genetically encoded reporters or fluorescent dyes is limited by the slow kinetics of the indicators. Signaling with axons beneath the compacted layers of the myelin sheath is inaccessible by electrophysiological and live-cell imaging methods, which are obscured by the thick layers of the compacted myelin membrane. Mathematical modeling of the kinetics of local and global signaling in activity-dependent myelin plasticity is hence an important tool for guiding the determination of the types of inter- and intracellular signaling molecules involved in the mechanisms of myelin plasticity.

Confocal imaging of local calcium responses in myelinating oligodendrocytes in cell culture, together with the imaging of local translation of myelin basic protein, show that action potentials in axons cause local calcium transients in OL via vesicular release of glutamate from axons acting on NMDA and glutamate receptors (mGluR) on the oligodendrocyte processes *Wake et al., 2011*. This promotes the formation of an axo-glial signaling complex, triggering the local translation of myelin basic protein to initiate myelination *Wake et al., 2011*, preferentially on the electrically active axon *Wake et al., 2015*. This local signaling can trigger myelin synthesis rapidly, within minutes *Wake et al., 2011*. The latency of local calcium signaling in OL was slower than the 80 ms required by our model but faster than 500 ms (Figure S7 in *Wake et al., 2011*). This reflects the slow kinetics of GCaMP2 calcium indicator and the actual signaling kinetics is likely much faster, not occurring via synaptic vesicles but rather by the action-potential-induced exocytosis on glutamatergic vesicles at axon varicosities *Wake et al., 2015*.

Similar responses are observed using in vivo imaging in zebrafish *Hines et al., 2015*; *Mensch et al., 2015*; *Krasnow et al., 2018* which further indicate that myelin sheath elongation during development is regulated by the kinetics of calcium transients in oligodendrocytes that are evoked by neuronal activity. In *Krasnow et al., 2018* they show that local calcium transients in oligodendrocyte cell processes are independent from one another and that the local calcium signals can be integrated within the cell to trigger global calcium responses in the cell body via temporal summation (Supp Figure 1 in *Krasnow et al., 2018*). The same study shows that myelin sheath elongation is promoted by high-frequency calcium transients, and sheath shortening is associated with low-frequency calcium transients. It also shows that the elongation occurs approximately one hour after Ca2+ while sheath shortening happens on a much longer time scale. Note the similar asymmetry in the OMP model between the myelin addition and removal rates ($\lambda_A$ and $\lambda_R$) but with the important difference that the myelin removal in our model does not depend on activity directly and is controlled only homeostatically. This further emphasizes the need to develop an activity-dependent mechanism of myelin homeostasis in future OMP models.

In spite of accumulating evidence, our suggestion that the role of $G$ might be played by calcium is only speculative and requires further investigation. There are many mechanisms of axon-OL interactions and a recent review of those can be found in *Munyeshyaka and Fields, 2022*. The OLs express a wide range of calcium channels that can regulate OL formation and function, and the diverse roles they play have already been investigated *Paez and Lyons, 2020*; *Wake et al., 2011*; *Wake et al., 2015*; *Fields, 2015*; *Krasnow et al., 2018*. For example, in *Wake et al., 2011* it was shown that the local calcium transients, released via glutamatergic vesicles and in response to action potentials firing in OL processes, can be blocked by botulinum toxin, but a global somatic calcium response persists, due to purinergic receptors that are expressed throughout the OL cell membrane. It is also important to note that oligodendrocyte progenitor cells (OPCs) often couple synaptically to axons via both, the excitatory, glutamatergic *Kukley et al., 2007*; *Ziskin et al., 2007* and inhibitory, GABAergic connections *Maldonado and Angulo, 2015*. This enables the cells to respond to different patterns of action potentials with different functional effects on cell differentiation and proliferation *Nagy et al., 2017*, for example, by increasing the number of OL available in the OC. However, while the OPCs can play an important role in modifying myelin content they do not have the needed feedback for adjusting the CV in a manner that eventually leads to a synchronized arrival of spikes at axonal targets. There are other activity-dependent signaling molecules released from axons firing action potentials, notably ATP and adenosine, that activate purinergic receptors, leading to global increases in cell calcium and activation of the myelination promoting genes *Stevens et al., 2002*; *Ishibashi et al., 2006*; *Fields and Ni, 2010*. The mechanisms by which gene expression can exert local changes in a given OL process are complex, as is the case for synaptic modifications, hence suggesting the identity of the reactions and reactants that play the role of the factor $M$ in our model is outside the scope of the present work.

## Materials and methods
### OL equation and spike-response curves
The spike response curve, $R(t)$, represents the global response of an OL to a single neuronal spike on any of its myelinated axons and gives OMP its timing-dependent character. In a more general OMP model, there can be several such curves responding to any triggering event in a cascade of responses, as depicted in *Figure 1—figure supplement 1*. Each of the responses represents the change in the concentration or amplitude of some global factor/signal released in the OL after each spike. A general spike response curve is parameterized by the separate rise and decay times, $\tau_r$ and $\tau_d$; for a spike occurring at time $t_s$, it can be written as,

$$R(t, \tau_r, \tau_d, Q \mid t_s = 0) = Q \frac{\tau_r + \tau_d}{\tau_d^2} e^{-t/\tau_d} \left(1 - e^{-t/\tau_r}\right),$$ (5)

where $Q$, represents the single release amount/quantity, which is constant and independent of the current value of a global signal, $G(t)$. The peak of the response is happening at time $t_{\max} = \tau_r \log \frac{\tau_r + \tau_d}{\tau_r}$, reaching the value $G_{\max}(t) = \frac{Q}{\tau_d} \left(\frac{\tau_r + \tau_d}{\tau_r}\right)^{-\frac{\tau_r}{\tau_d}}$. This form allows more general explorations (e.g., $\tau_r \approx 0$, yields the exponentially decaying $R(t)$), and most importantly it is the impulse response curve of the second-order linear system,

$$\tau_d \tau_r \ddot{G}(t) + \left(\tau_d + 2\tau_r\right) \dot{G}(t) + \frac{\left(\tau_d + \tau_r\right)}{\tau_d} G(t) = s(t),$$ (6)

where $s(t)$ represents the signal that drives the system, and $\dot{G}(t)$ and $\ddot{G}(t)$ are the first and second time derivatives of $G(t)$.

In order to simplify extensive explorations of all OMP parameters, here we chose a simplified form for $R(t)$ that uses a single characteristic time of the OL spike response, $\tau_G = \tau_r = \tau_d$, yielding,

$$R(t) = \begin{cases} \frac{2}{\tau_G} e^{-t/\tau_G} \left(1 - e^{-t/\tau_G}\right), & t > 0 \\ 0, & \text{otherwise.} \end{cases}$$ (7)

In *Figure 1E*, we show a set of such curves for varying values of $\tau_G$. With this simplification, the differential equation for $G(t)$, corresponding to *Equation 6*, becomes

$$\tau_G^2 \ddot{G}(t) + 3\tau_G \dot{G}(t) + 2G(t) = s(t), \tag{8}$$

which we use to simulate the global response of an OL to input $s(t)$.

OMP is a continuous time model and the input signal, $s(t)$, appearing in *Equation 8*, can technically be any integrable input. However, the time-dependent MP will have to rely on events that are sharply defined in time. We use trains of neuronal spikes for individual axons which are prescribed using a particular interspike interval (ISI) distribution, $p_{ISI}(t_{(i)})$, that is, they are generated with a renewal process. An important parameter that characterizes these trains is their mean firing rate, $f_s$, or equivalently the mean ISI, $\tau_s = \langle t_{(i)} \rangle_t = 1/f_s$. We use two main forms for $p_{ISI}$ distributions: (a) the exponential distribution with the rate $f_s = 1/\tau_s$ with added constant refractory time, $t_R$, yielding the refractory Poisson process and (b) regular spiking, spaced at constant intervals, $\tau_s$. These $p_{ISI}$ s cover two extremes: for Poisson spiking, with $t_R = 0$, the appearance of the next spike is completely independent of the previous spikes (memoryless process), and for regular spiking, the appearance of the next spike is precisely determined by the last spike. To both of the forms of $p_{ISI}$ we also add *jitter*, specified by $\sigma_j$, which spreads the spike times, such that these temporal shifts are normally distributed according to $\mathcal{N}(0, \sigma_j)$. Such a mix of refractory Poisson and regular spiking with added jitter seems to cover the spike dynamics for communication between many areas of the brain *Maimon and Assad, 2009*.

When the spikes on different axons follow the same renewal process, that is, obey the same $p_{ISI}$ distribution, we call these 'pure' signals, and among them distinguish two different cases: (1) the renewal processes for different axons are fully independent, and (2) they are time-locked via imposed relative time shifts between spikes in different axons, that is the fixed delays. Due to added jitter, which is always independent between different axons, the spikes will not be precisely time-locked, but will still be 'correlated'. In *Figure 2B*, we show examples of correlated and independent spike trains, prior to the imposition of fixed delays, for both, Poisson and regular spiking. Mathematically, spike trains are generally formulated as a sequence of Dirac delta functions,

$$s_a(t) = \sum_{k_a} \delta(t - t_{k_a}), \tag{9}$$

where $t_{k_a}$ indicates the time of the $k^{th}$ spike on axon $a$, and the sum goes over all spikes that have occurred prior to time $t$. When *Equation 9* is applied to the linear system in *Equation 8* the analytical solution for $G(t)$ becomes the sum of the responses to spikes on all axons that it myelinates, i.e.,

$$G(t) = \sum_a \sum_{t_{k_a} < t} R(t - t_{k_a}), \tag{10}$$

which can also be viewed as the sampling of $R(t)$ in reverse time, which we use in our theoretical derivations (see *Figure 1F*). We do not use *Equation 10* directly, but rather solve *Equation 8* numerically, as described in the OMP Implementation and Simulations section.

## OMP model learning equations

The fluctuating global signal, $G(t)$, obtained via *Equation 8*, serves as a catalyst for the local myelin-promoting factor, $M$. We model this by making the increase in $M$ proportional to $G(t)$, as well as to the signal strength in the axon it myelinates. Since $M$ is also continuously converted to myelin with some rate, $\lambda_A$, the differential equation for its concentration on axon $a$, $M_a$, can be written as,

$$\dot{M}_a + \lambda_A M_a(t) = \lambda_M G(t) s_a(t), \tag{11}$$

where $\lambda_M$ specifies the production rate of $M$ (*Figure 1D*, *Table 1*) and it is the same for all processes. It is an OMP parameter that effectively controls the rate of adaptive changes in our model. The presence of the factor $M$ in a given axon and its effects will lead to the increase in myelin sheath thickness (with rate $\lambda_A$) and will compete with another continuous process of myelin removal which, in the absence of any activity, decreases with some rate, $\lambda_R$.

For most neuronal plasticity models, saturation functions need to be introduced to stabilize the learning process. Similarly, in our implementation of OMP, we introduced two separate saturation functions for myelin addition, $F_s^A$, and removal, $F_s^R$. The OMP equation for the latency on axon $a$, $\tau_a$, can be written as

$$\dot{\tau}_a = \lambda_R F_s^R(\tau_a(t)) - \lambda_A F_s^A(\tau_a(t))M_a(t), \tag{12}$$

where $F_s^R(\tau) = H_r(\tau_{\max} - \tau)$ and $F_s^A(\tau) = H_r(\tau - \tau_{\min})$ , $H_r(x) = xH(x)/(\tau_{\max} - \tau_{\min})$ is the normalized ramp function, $H(x)$ is the Heaviside (unit step) function, $\tau_{\max}$ and $\tau_{\min}$ are the parameters of the model which specify the maximal and minimal delays that are attainable on any axonal connection, respectively.

We make two modifications to this basic model, one being the case of $\lambda_A \to \infty$ (*instantaneous myelination*), for which **Equation 11** is not needed, and another case includes a homeostatic equation (**Equation 17**), which presumes that overall myelination for each OL reflects a long-term homeostatic steady-state between adding and removing myelin.

## 'Instantaneous' Myelination

The conversion rate, $\lambda_A$, appears not to play an important role (**Figure 4—figure supplement 2D**), particularly if OLs operate far from the saturation limits, as $M$ is then just a currency for conversion into myelin, or, changes in CV. We can eliminate **Equation 11** by taking the limit $\lambda_A \to \infty$, that is, presume that $M$ is instantly converted to myelin, resulting in an immediate change in the time delay, $\tau_a$. The solution for $M_a(t)$ is obtained by convolving the impulse response of the left side of **Equation 11**, $e^{-\lambda_A t}$, with the expression on the right side, i.e.,

$$M_a(t) = \lambda_M \int_0^t e^{-\lambda_A(t-t')} G(t')s_a(t')dt' + M_a(0)e^{-\lambda_A t}. \tag{13}$$

Replacing **Equation 13** in **Equation 12** and using the fact that $s_a(t)$ is a spike train we obtain

$$\dot{\tau}_a = \lambda_R F_s^R(\tau_a(t)) - \lambda_M \sum_{t_{k_a} < t} G(t_{k_a})F_s^A(\tau_a(t))\lambda_A e^{-\lambda_A(t-t_{k_a})}. \tag{14}$$

The expression $f_{\lambda_A}(t) = \lambda_A e^{-\lambda_A t}$, appearing in **Equation 14**, can be interpreted as a Dirac delta function when $\lambda_A \to \infty$, since $\lim_{\lambda_A \to \infty} \int_{-\infty}^{\infty} g(t)f_{\lambda_A}(t-a)dt = g(a)$. The 'instantaneous' equivalent of **Equation 12** can then be written as

$$\dot{\tau}_a = \lambda_R F_s^R(\tau_a(t)) - \lambda_M \sum_{t_{k_a} < t} F_s^A(\tau_a(t_{k_a}))G(t_{k_a})\delta(t - t_{k_a}), \tag{15}$$

where it is indicated that $\tau_a(t)$ in the sum will only depend on its values at spike times $t_{k_a}$, after **Equation 15** is integrated. To have stable integration in this case, it is important to set $\lambda_M$ sufficiently small, particularly when used with homeostatic regulation described below.

## Homeostatic Regulation

The form in **Equation 15** illustrates that the essence of the adaptive process for timing adjustments is the balance between continuous myelin removal (longer delay) and the discrete increments induced by spikes. For independent Poisson spikes and ignoring saturation effects, the balance condition is,

$$\lambda_R = \lambda_M N_A / \tau_s^2, \tag{16}$$

where $\tau_s$ is the average inter-spike interval of the Poisson process on a single axon. However, with the saturation functions and when correlated signals are introduced, this homeostatic balance can be disturbed. In order to keep the system in balance, we make the removal rate of myelin, $\lambda_R$, another time dependent variable in the dynamics of the OMP model. To do so, we assume that each OL operates with some local and nominal homeostatic level of myelination, parameterized by some nominal delay, $\tau_{\mathrm{nom}}$, such that any deviation from it will lead to a slow change in $\lambda_R$ according to,

$$\dot{\lambda_R} = \lambda_H \lambda_R(t)(\tau_{\mathrm{nom}} - \frac{1}{N_A}\sum_{a=1}^{N_A} \tau_a) \tag{17}$$

where $\lambda_H$ is a homeostatic rate. We set it to a very small value ($\lambda_H \leq 10^{-5}$) so that the time scale for changes in $\lambda_R$ is much slower than the time scale of individual spikes or the changes in myelination. In some instances, in order to test the importance of having **Equation 17**, we simply set $\lambda_H$

to 0, after guessing or finding the value for $\lambda_R$ that balances the increase in myelin content for a given input signal. This homeostatic rule can be interpreted as a tendency of each OL to conserve its overall amount of myelin, while re-distributing it over different axons. We assume that such an activity-dependent MP process is in place and simulate only its ability to adjust the overall rate of myelin removal, that is the myelin removal rate, $\lambda_R$.

## OL Chain

The consistency of *Equation 4* can, for small jitter and fixed delays, cause instabilities as the 'leading' axon will keep losing myelin more than other axons. Generally, the spikes on less delayed axons will produce less $M_a$ than spikes on more delayed ones, creating an imbalance. This trend will stop only after the rate of change is significantly slowed down by the saturation limits and the decreasing $\lambda_R$, due to homeostatic regulation. This problem is not surprising as most of the plasticity models are inherently unstable (e.g. Hebbian) and rely on saturation mechanisms. In the case of OMP models, this problem can also be resolved by using a natural assumption that the final arrival time will depend on the action of all OLs along a given axon bundle; the temporal differences of spikes across the bundle will become smaller for OLs close to the target compared to OLs close to the sources. For this reason, it is worth, if not necessary, to simulate the sequential action of multiple OLs, in which the preceding OL-axon bundle segment can pass its modified spike arrival times to the next segment. Hence, we simulate a sequence of OMP equations, each feeding its output to the next segment in the OC (see *Figure 2A*). The OC will have $N_O$ sets of OL equations, each having its own myelin-promoting factors, $M_a$, and its own local delays, $\tau_a^{(o)}$. As already emphasized, the OC depicted in *Figure 2A* does not imply literally that there are $N_O$ OL cells along the axons, but rather that there are $N_O$ segments, representing $N_O$ different populations of oligodendrocyte cells myelinating different portions of the axonal bundle, which modulate the delays locally. Assuming that all cells within the same segment will receive the same pattern of spikes and respond to it in the same way, they all can be governed by a single OMP equation. Individual oligodendrocyte cells, in fact, would not be able to modify the delays effectively and independently from other oligodendrocyte cells in the same location, as they would not be able to form tight nodes of Ranvier, considering that OLs prefer not to myelinate the same axon multiple times. Neighboring OL cells are then needed to stack their processes, reducing the width of the nodes of Ranvier and in this way greatly increasing the CV, that is, reducing the conduction delays. Hence, we have a sequence/chain of $N_O$ "effective" OL cells, each modifying its local fraction of the total delay along $a^{th}$ axon, $\tau_a^{(o)}$, so that a total conduction delay on the axon is just the sum of all local delays $\tau_a = \sum_o \tau_a^{(o)}$.

## OL-axon Connectivity

In general, each OL myelinates many axons and can have multiple processes on a single axon (*Figure 1C*), or can have none on others. The OL-axon connectivity can be mathematically described with the *myelination matrix*, $\mathcal{M}$, with OLs as rows and axons as columns, and indicates the number of processes a given OL places on each of the axons, for example, for the 'general' connectivity in *Figure 1C*, the matrix is,

$$\mathcal{M} = \begin{bmatrix} 3 & 2 & 4 & 2 \\ 2 & 1 & 2 & 2 \\ 0 & 3 & 0 & 2 \end{bmatrix}, \tag{18}$$

while the observed avoidance of OLs myelinating single axons with multiple processes *Dumas et al., 2015*; *Walsh et al., 2016*, makes $\mathcal{M}$ likely to consist of ones and zeros. Here, we use an 'effective' OL, which represents a population of OLs behaving in an identical manner, and hence we also treat the OL-axon connectivity in an 'effective' way. We currently only address a simple situation in which the OC contains $N_O$ OLs that are fully myelinating a given bundle of $N_A$ axons, that is, $\mathcal{M}$ is simply an $N_O \times N_A$ matrix of ones. We assume stable connectivity and that modifications of CV are achieved only through the remodeling of myelin sheaths and the nodes of Ranvier. The issue of precise, cellular-level *OL-axon connectivity* becomes important in more detailed OMP models in which individual OLs are simulated and in which combined effects on the CV of multiple OLs myelinating the same location along an axon can be addressed; however, such models will be computationally

extremely demanding. The issues of random or partial connectivity (the OLs in the chain myelinating different subsets of the $N_A$ axons considered) can still be addressed and we expect those to have only less effective but not detrimental synchronization effects. The lower efficiency when using more realistic $\mathcal{M}$ might not be an issue, considering that the number of OLs in the OC that we simulate is vastly lower than the number of actual cells myelinating axons along a given pathway.

## OMP implementation and simulations

In its fullest form, the OMP model can have up to 13 scalar parameters plus the $N_O \times N_A$ myelination matrix $\mathcal{M}$. We provide an overview of all OMP parameters in *Table 1*, together with the OMP variables and other symbols used for specifying the spiking signals conducted along the axons and for the data analysis.

The nominal OMP model is governed by three main *Equations 8; 11; 12*, expanded with *Equation 17*, which adds a homeostatic control of $\lambda_R$, that now becomes an OMP variable. Omitting the equation for homeostasis requires fine-tuning the balancing value for $\lambda_R$ for a given parameter setting. This model can be simplified further by assuming 'instantaneous myelination', in which case *Equation 11* can be omitted, when using *Equation 15* instead of *Equation 12*; however, using *Equation 15* requires, particularly if used together with *Equation 17*, that $\lambda_M$ be chosen sufficiently small to insure that the integration is stable. If $\lambda_M$ is large, sudden jumps in the value of $\tau_a$ will make *Equation 17* stochastic. This can lead to negative values for $\lambda_R$ or even the delays themselves, which is not realistic.

The standard variables of the model, for a single OL, are $G(t)$, $M_a(t)$, and $\tau_a(t)$, $a = 1 \ldots N_A$, giving a total of $2N_A + 1$ variables per OL, excluding $\lambda_R$. In practice, solving the model will require $2N_A + 2$ variables per OL, since *Equation 8* is a $2^{nd}$-order differential equation, requiring an auxiliary variable (see the next section). $G(t)$ and $M_a$ are 'fast' variables that change on a milliseconds time scale (dictated by $\tau_G$ and impulse responses to the spikes, with mean ISI, $\tau_s$), while, $\lambda_R$ and $\tau_a$, $a = 1 \ldots N_A$, are 'slow' variables whose rate of change is controlled by $\lambda_H$ and $\lambda_A$, respectively ($\lambda_H \ll \lambda_A$). In *Figure 1—figure supplement 2A, B*, we show an example of time progression for both fast and slow variables, as well as the synchrony measure, $\sigma_\tau$ (*Figure 1—figure supplement 2B*, top row) and the epoch-averages of $M_a$ (bottom row), when a set of correlated signals is conducted along OC. Examples shown are for an OL at the beginning of the oligo-chain with $N_O = 5$, $N_A = 10$, and a single trial, except for $\sigma_\tau$ (dashed lines) and $\lambda_R$ (different colors), where the results from independent trials (3 total) are also shown. In *Figure 4—figure supplement 3* we similarly show the time-progression for the unstable case, $N_O = 1$, as shown in *Figure 3A* ($\lambda_H = 10^{-6}$), but now for three different values of $\lambda_H$, indicating that the oscillations seen in $\sigma_\tau(t)$ are a result of homeostatic control. We note here, again, that $N_O = 1$ does not mean literally that there is a single oligodendrocyte acting on a given axonal bundle but rather a population of oligodendrocytes acting at a particular location along the axon receiving the same pattern of activations and responding to it in the same way. When independent signals are conducted along OC, the time progression of the slow OMP variables displayed only stochastic variations and no clear trends, as shown in *Figure 2—figure supplement 2* (apart from initial 'refocusing' of all $\tau_a$ to the same mean value, due to homeostatic constraints). This result was consistent across all simulations in which independent signals are used, with either Poisson or regular spiking $p_{ISI}$.

### Implementation of the OMP model

The crucial element of our model implementation is solving a system of differential *Equations 8; 11; 12*. To solve *Equation 8* we implemented a more general case, given by *Equation 6*, which can be written as a set of first-order equations using an auxiliary variable, $v(t)$,

$$
\begin{aligned}
\dot{v}(t) &= -a\, b\, G(t) - (a+b)v(t) + q_s \ > \Sigma_{t_k}\delta(t - t_k), \\
\dot{G}(t) &= v(t), \ \text{with } G(0) = 0,
\end{aligned}
\tag{19}
$$

where constants, $a$, $b$, and $q_s$, are defined using the parameters of the general spike response curve (*Equation 5*, i.e., the characteristic rise, $\tau_r$, and decay, $\tau_d$, time constants, and the amplitude of the release, $Q$) as follows

$$
a = \frac{\tau_r + \tau_d}{\tau_r \tau_d}, \ b = 1/\tau_d, \ q_s = \frac{Q(\tau_r + \tau_d)}{\tau_r \tau_d^2}.
$$

To use *Equation 8* with single characteristic time, for simplicity, as is described in this manuscript, we set the characteristic rise and decay times to be equal, i.e., $\tau_r = \tau_d = \tau_G$ ($t_{\max} = \tau_G \ln 2$). We also set $Q = 1$ for all spikes, in which case the constants in *Equation 19* simplify to $a = 2/\tau_G, b = 1/\tau_G, q_s = 2/\tau_G^2$. Implementation of *Equation 11* is straightforward, as it just adds one more first-order differential equation, while implementation of *Equation 12* requires special handling of Dirac delta functions. The simplest way to do this is to integrate the OMP equations between subsequent spikes, as the spikes are specified externally, and handle the discontinuities at each spike separately. For more sophisticated/alternative OMP schemes, which might contain a cascade of triggered events, as depicted in *Figure 1D*, thus, requiring some kind of "events" functionality (functions evaluated when a set of conditions on the variables are satisfied), which is available in many integration packages. *Van Rossum and Drake, 1995* solver in *Virtanen et al., 2020*, *scipy.integrate.solve_ivp*, and the full simulation code, including the specification of parameters as well as subsequent data analysis, was implemented in Python. Depending on the parameters used, simulations took anywhere between a few hours to more than 10 days (including the repeats, $n_r \in [3, 10]$). Implementation of the same code in C, or using JIT *Ansmann, 2018*, could make evaluations substantially faster, but might sacrifice some flexibility and ease in implementing the event handling.

## OMP model evaluation

We conducted our simulations on a highly parallel National Institutes of Health Biowulf cluster (http://hpc.nih.gov) and divided them into more than a dozen studies. For each simulation study, we specify a set of values to be explored for each of the OMP parameters, which are listed in Appendix 1. In the early and exploratory phase, we coarsely identified the working range for the most important parameters and usually chose a few values, usually three, indicating the low, mid, and high values of its "working" range, and include sometimes explorations outside that range (as was done in our early, coarse exploration of the model). Due to a large number of parameters that could influence the performance, we could not afford to exhaustively explore all of them on a fine grid for a large range of values. Instead, in each study, we explored the influence of a particular parameter on a finer grid, or in some cases a group of parameters, for example, the exploration of $\tau_G$ and $\tau_s$, shown in *Figure 3E* (see the right panel in *Appendix 1—table 1*).

In our simulations, we chose fixed delays to be randomly drawn from a normal distribution $\mathcal{N}(D_{\mathrm{mean}}, \sigma_D)$. Here $\sigma_D$ is the standard deviation among fixed delays $t_d^{(a)}$ on all axons and is an important parameter in our simulations, while $D_{\mathrm{mean}}$ is an arbitrary offset, insuring that the delays are positive. This positivity constraint is not very important since we were only interested in the spread of the synchronized spikes on different axons. These fixed delays are always combined with adaptive delays, that is, the conduction delays, $\tau_a$, on myelinated axons, so we often normalize them to zero mean ($D_{\mathrm{mean}} = 0$), and we label such normalized fixed delay on axon $a$ as $D_a$. The spread of arrival times of synchronized spikes at the target will then simply be the standard deviation of $D_a + \tau_a$, that is, $\sigma_\tau = \mathrm{SD}(D_a + \tau_a)$. We use $\sigma_\tau$ extensively in this manuscript, as a measure of such spread in arrival times, making it an inverse measure of synchronization with $\sigma_\tau = 0$ indicating perfect synchronization. In order to allow for easier comparison between different sets of parameters, particularly when plotting the average $\sigma_\tau$ over many runs (see *Figure 3—figure supplement 1A* and the left panel in C), we introduce a normalization step, so that the initial spread due to fixed delays is exactly $\sigma_D$, i.e., $D_a^{\mathrm{norm}} = \sigma_D D_a/\mathrm{SD}(D_a)$.

We initialized a set of $N_A$ random fixed delays, parameterized by their spread, $\sigma_D$, for each trial ($n_r$ total). We initialized the two $N_O \times N_A$ matrices, one carrying the information about $M$ concentration for each OL process in OC, and the other carries the local delays on each axon for each OL in the OC. The former is initialized to zero (all $M_a = 0$) and the latter is initialized based on the specified mean, $\tau_0$ and the percent spread, $p_\tau$, i.e., $\tau_a \sim \mathcal{N}(\tau, p_\tau \tau/100)$. For simplicity, we set $\tau_0 = \tau_{\mathrm{nom}}^o = \tau_{\mathrm{nom}}/N_O$, and we used $p_\tau = 5\,\%$. We created $N_A$ spike trains with prescribed dynamics, based on a given $p_{\mathrm{ISI}}$, and parameterized by $\tau_s$ (examples shown in *Figure 2B*). For time-locked signals, we generated a single train with a given $p_{\mathrm{ISI}}$, and others were shifted versions of the same, based on the values of fixed delays ($D_a$, or $t_d^{(a)}$), which are subsequently randomized using jitter. The jitter spreads the location of all spikes, such that these temporal deviations are normally distributed according to $\mathcal{N}(0, \sigma_j)$. For each trial, we ran $n_e$ epochs of learning, each with duration $T_e$, and collected the values of the 'slow' variables, $\sigma_\tau, \lambda_R$, as well as the mean value during the epoch of $\tau_a$ and $M_a$, for every OL in the OC.

Due to the large number of runs we conducted, only $\sigma_\tau$ information was saved for every run, while other variables were saved only if the analysis required it. When the homeostatic equation was used ($\lambda_H > 0$), we allowed 1 or 2 extra ('warm up') epochs to run, during which modifications to $\tau_a$ were disabled, allowing $\lambda_R$ to be closer to its equilibrium value when we start to track the spread, $\sigma_\tau$. Due to oscillatory behavior of $\lambda_R$, in most cases, this was not very important to do and did not affect $\sigma_\tau(t)$.

## Model fitting for synchronization profiles

The averages over a large number of runs can obscure the details in synchronization profiles, $\sigma_\tau(t)$, obtained from single runs, such as oscillations and other instabilities. Hence, it is useful to summarize and condense thousands of obtained profiles in an automated fashion and obtain distributions of critical parameters, most importantly the long-time baseline parameter, $\sigma_\tau^\infty$, and the 'learning time', $\tau_L$, which is the inverse of the learning rate, $L_\tau$. We do this by fitting a sufficiently rich model, with additional parameters, $p_i$, that is able to capture most of those profiles reasonably well. The general form of the full (unrestricted) model we use can be written as,

$$\sigma_\tau(t) \sim \sigma_\tau^\infty + (p_3 \exp(-t/\tau_L) - p_4 \exp(-t/p_5))f_1(t) + f_0(t), \tag{20}$$

where the damped oscillatory behavior of $\sigma_\tau(t)$ is described via functions $f_0$ and $f_1$ (with long-time limit 0 and 1, respectively), used to quantify the observed instabilities in learning. For $f_0(t)$ and $f_1(t)$, we have used cosine functions with additional parameters describing their amplitude, period, and phase, as well as their damping factor. Hence, the most general fitting model we use, containing 12 parameters, is a double exponential function containing both, the multiplicative and damped additive cosine oscillation. We label the full model as E2C2 and its formula is

$$\sigma_\tau(t) \sim \sigma_\tau^\infty + (p_3 \exp(-t/\tau_L) - p_4 \exp(-t/p_5))(1 + p_6 \cos(2\pi t/p_7 + p_8)$$
$$+ p_9 \exp(-t/p_{10}) \cos(2\pi t/p_{11} + p_{12})) \tag{21}$$

We then explore a set of restricted models, all nested within the full model above, starting with the simplest, constant model (C), having only one free parameter, $\sigma_\tau^\infty$, which, consequently is also contained in all other models, with progressively more parameters. These are: single exponential model (E1), double-exponential (E2), double-exponential with multiplicative oscillation (E2C). The full set of models that we fit is then,

- C: $\sigma_\tau(t) \sim \sigma_\tau^\infty$
- E1: $\sigma_\tau(t) \sim \sigma_\tau^\infty + p_3 \exp(-t/\tau_L)$
- E2: $\sigma_\tau(t) \sim \sigma_\tau^\infty + (p_3 \exp(-t/\tau_L) - p_4 \exp(-t/p_5))$
- E2C: $\sigma_\tau(t) \sim \sigma_\tau^\infty + (p_3 \exp(-t/\tau_L) - p_4 \exp(-t/p_5))(1 + p_6 \cos(2\pi t/p_7 + p_8))$
- E2C2: full, unrestricted model, shown in *Equation 21*.

Our $\sigma_\tau(t)$ consists of $n_{\text{pts}} = n_e$ data points which we use to estimate parameters for all models, via independent fits. We constrained the parameters during the fitting as follows: $0 \le \sigma_\tau^\infty \le 2\sigma_\tau^{(0)}$, $T_{\text{exp}}/1000 < \tau_L < 1000 * T_{\text{exp}}$, $0 < p_3 < 2 * \sigma_\tau^{(0)}$, $0 < p_4 < 2 * \sigma_\tau^{(0)}$, $T_{\text{exp}}/1000 < p_5 < 5 * T_{\text{exp}}$, $0 < p_6 < \sigma_\tau^{(0)}/2$, $T_{\text{exp}}/25 < p_7 < \infty$, $0 < p_8 < 2\pi$, We used these limits to have better stability and to avoid extreme outliers (since the total number of runs was close to $100k$, we did not inspect every single fit, but only a small fraction of all fitted parameters values were obtained at the boundary). An example of such fits is shown in *Figure 3—figure supplement 2A*. We conducted a large number of randomly initialized fits, in order to insure that the best possible fit is obtained (*Figure 3—figure supplement 2A, B*). Due to the complexity of the unrestricted model, E2C2 might not end up finding the true global minimum and having the lowest MSE, but that happened very rarely (<0.3% or runs, and in all those cases E2C was the one with the minimal MSE). In most cases, different fitting models give very similar estimates for the most important parameter, $\sigma_\tau^\infty$, but in many cases the estimates can differ substantially, depending on what model is selected (see *Figure 3—figure supplement 2A, B*). Our desire is to opt for a simpler model, when possible, as it often provides more reliable estimates of $\sigma_\tau^\infty$ and also $\tau_L$ (unless it is model C, for which $\tau_L \to \infty$). This allowed us to categorize better the behavior of OMP and particularly to identify single runs in which the approach to synchrony was truly unstable, or oscillatory, or if $\sigma_\tau$ was diverging away from synchrony. This categorization was implemented by our model selection procedure described below.

## Model selection

All of the simpler/nested models we call restricted since they are essentially equivalent to the unrestricted model with the coefficients of all extra explanatory variables being restricted to zero. Of course, the unrestricted model, having more parameters will then always be able to fit the data better or at least as well as the restricted model, in terms of the mean-squared error (MSE). The question is, whether this improvement is sufficiently large to warrant sacrificing the level of parsimony of the restricted model. One approach to this problem is to use an F-test, which compares two models, the unrestricted (U) vs a particular restricted version (R). Given the obtained fits to $n_{pts}$ points for both models and their corresponding residual sum of squares (RSS) one can calculate the F-statistic, $F$, given by

$$F_{\text{num}} = \frac{\text{RSS}_R - \text{RSS}_U}{n_U - n_R}$$

$$F_{\text{den}} = \frac{\text{RSS}_U}{n_{\text{pts}} - n_U}$$

$$F = \frac{F_{\text{num}}}{F_{\text{den}}}$$

where $\text{RSS}_m$ is the residual sum of squares of model $m$. This $F$ is $F$-distributed, with $(n_U - n_R, n_{\text{pts}} - n_U)$ degrees of freedom, which we can use for our statistical tests. The null hypothesis in these tests is that the unrestricted model does not provide a significantly better fit than the restricted model, hence rejecting it at a given significance level means that a more complicated model is needed.

Using the conventional F-test for model selection did not work well for our purposes, since the undulations we observe are nearly always statistically significant even if the most stringent tests with extreme significance levels were used, for example, $p < 10^{-15}$. This was not surprising, as the synchronization profiles were never truly statistically constant or exponential. For example, in *Figure 3— figure supplement 2C*, when $\sigma_\tau(t)$ is observed on a full scale, one would expect that the constant model is the best description of the behavior observed. However, the model selection with regular F-test chooses the less restricted models, in fact E2C2 in this case. After zooming in, we see that the results were indeed not constant, as those deviations from constancy were not just due to noise, but were significant. If the obtained simulation profiles were noisier then the restricted models would have a reasonable chance of being selected. Instead of adding artificial noise to our $\sigma_\tau(t)$, we introduced a parameter, $p_{\text{MSE}}$, with which we essentially control what level of noise or deviations we deem tolerable. The $p_{\text{MSE}}$ expresses this level of tolerance as the percentage of the initial, $\sigma_\tau$, or approximately, $\sigma_D$. Hence, we declare the minimal amount of RSS in any fit, $\text{RSS}_{\text{min}} = n_{\text{pts}} * \text{MSE}_{\text{min}}$ and $\text{MSE}_{\text{min}} = (p_{\text{MSE}} * \sigma_\tau^{(0)}/100)^2$. This sets the level of MSE that is presumed by default, that is, some minimal amount of noise present in residuals of any model. This is essentially specifying how much of RMS error can be tolerated in the restricted model, in order to reject the null hypothesis that the unrestricted model is better. Since the numerator will remain unchanged, this essentially only modifies the denominator,

$$F_{\text{den}}^m = \frac{\text{RSS}_U + \text{RSS}_{\text{min}}}{n_{\text{pts}} - n_U},$$

where $\text{RSS}_{\text{min}} = n_{\text{pts}} * \text{MSE}_{\text{min}}$ and $\text{MSE}_{\text{min}} = (p_{\text{MSE}} * \sigma_\tau^{(0)}/100)^2$. The numerator portion is not affected, as it is a difference between two RSS. This yields the modified F-statistic, $F^m$,

$$F^m = \frac{F_{\text{num}}}{F_{\text{den}}^m} \tag{22}$$

that we use for our tests. Here we use the modified F-statistic (*Equation 22*, in most cases with $p_{\text{MSE}} = 2$ %). Under the modified test, the curve shown in *Figure 3—figure supplement 2C* is now declared as model C ("constant") for the three smallest significance levels used (see below).

In our procedure, we start with the model with minimal MSE (usually E2C2), test it against all of the restricted models, and choose the most restricted model for which the null hypothesis is not rejected. We performed the tests separately at different but very low significance levels ($\alpha \in [0.01, 0.00001, 10^{-10}, 10^{-15}]$). The choice of the model was in many cases not strongly

influenced by the choice of $\alpha$, but was strongly dependent on the choice $p_{\mathrm{MSE}}$. For $p_{\mathrm{MSE}} = 0$ %, even at extremely low $\alpha$, the unrestricted model would always be chosen (see *Figure 3—figure supplement 2C*). Using $p_{\mathrm{MSE}} = 2$ %, allowed the choice of model to depend on $\alpha$ in most cases, as is indicated in *Figure 3—figure supplement 2A*, where E2C2, E2C, or E1 would be chosen, depending on how stringent the test was. We chose $p_{\mathrm{MSE}} = 2$ %, based on a set of 100 random examples in which the best model is chosen manually by visual inspection (e.g., the constant model in *Figure 3—figure supplement 2C*). Note that our model selection is largely ad-hoc and we emphasize that our modified F-test does not aim to provide a quantitative statistical analysis, as use of such absurdly small significance levels indicates, but only to provide a useful quantitative tool for summarizing tens of thousands of runs that we have performed. While the distribution of different models changes significantly for different choices of $p_{\mathrm{MSE}}$ and $\alpha$, the derived values of the parameters $\sigma_\tau^\infty$ and $\tau_L$ is not significantly changed when different values of $p_{\mathrm{MSE}}$ (but >1%) and $\alpha$ (but smaller than 0.01) were used. The same holds for the ad-hoc rule, of reverting to a less restricted model when $\mathrm{MSE}_R > 500 \times \mathrm{MSE}_U$, which happened very infrequently (see *Figure 3—figure supplement 2D*).

## Acknowledgements

This research was supported by the Division of the Intramural Research Programs (DIRP) of the National Institute of Mental Health (NIMH), USA, (SP, DP, ZIAMH002797) and the Eunice Kennedy Shriver National Institute of Child Health and Human Development (NICHD), USA, (RDF, ZIAHD000713; PJB, 1ZIAHD008972-04). This work utilized the computational resources of the NIH HPC Biowulf cluster (http://hpc.nih.gov).

---

## Additional information

### Funding

| Funder | Grant reference number | Author |
|---|---|---|
| Intramural Research Program of NIMH/NIH | ZIAMH002797 | Sinisa Pajevic Dietmar Plenz |
| Intramural Research Program of NICHD/NIH | ZIAHD000713 | R Douglas Fields |
| Intramural Research Program of NICHD/NIH | 1ZIAHD008972-04 | Peter J Basser |

The funders had no role in study design, data collection and interpretation, or the decision to submit the work for publication.

### Author contributions

Sinisa Pajevic, Conceptualization, Resources, Data curation, Software, Formal analysis, Supervision, Validation, Investigation, Visualization, Methodology, Writing – original draft, Project administration, Writing – review and editing; Dietmar Plenz, Resources, Supervision, Funding acquisition, Visualization, Project administration, Writing – review and editing; Peter J Basser, Resources, Supervision, Project administration, Writing – review and editing; R Douglas Fields, Conceptualization, Supervision, Writing – review and editing

### Author ORCIDs

Sinisa Pajevic ⓘ http://orcid.org/0000-0002-8880-3320
Dietmar Plenz ⓘ http://orcid.org/0000-0002-0008-3657
Peter J Basser ⓘ http://orcid.org/0000-0003-4795-6088

### Decision letter and Author response

Decision letter https://doi.org/10.7554/eLife.81982.sa1
Author response https://doi.org/10.7554/eLife.81982.sa2

## Additional files

### Supplementary files
• Transparent reporting form

### Data availability
All results shown in our figures are produced via computer simulations of our model. The code and the scripts that generated these results are provided on GitHub https://github.com/pajevic/OMPmodel (copy archived at *Pajevic et al., 2023*). The only data shown that are not the result of our simulations are the images in panels A and B of Figure 1, which are reused with permission, as they also appeared in another publication.

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

# Appendix 1

## Tables of parameter values explored in simulations

The values explored for some OMP parameters were similar in different simulations and, to save space, in the tables below we only report the parameter values that differed from these most commonly used default values: $N_A = 10$, $N_O = 5$, $\tau_G = \{10, 20, 30\}$ ms, $\tau_s = \{50, 100, 200\}$ ms, $\lambda_M = \{0.01, 0.02, 0.05, 0.1\}$ ms$^{-1}$, $\lambda_A = \{0.1, 0.01\}$ ms$^{-1}$, $\lambda_H = 10^{-6}$ ms$^{-2}$, $T_e = 10$ sec, $n_r = 4$ and $n_e = 100$, $\sigma_D = \{5, 10\}$ ms, $\sigma_j = \{1, 3, 5\}$ ms, $\sigma_s = 0$ %, $\tau_{\max} = 100$ ms, $\tau_{\min} = 3$ ms, and $\tau_{\text{nom}} = 50$ ms, $t_R = 0$ ms. In two cases the differences with the defaults were small, so we report them only here in the text. The first is studying the influence of the number of axons, $N_A$, for which $\lambda_A = 0.01$, $N_O = \{5, 10\}$, and $N_A = \{5, 10, 20, 30, \ldots, 70, 80, 100, 150\}$ ($n = 4752$ runs). Runs with $\sigma_D = 5$ ms ($n = 2376$) are used in creating *Figure 3F*. The second explores the influence of $\sigma_D$ parameter, i.e., fixed delays, shown in *Figure 4F*, for which $\sigma_D = \{2, 3, 5, 7, 10, 15, 20\}$ ms and $n_r = 5$ ($n = 1512$). Below we show the tables for the other 8 sets, where the same units are used as above.

**Appendix 1—table 1.** Parameter sets used in our OMP model simulations.
Parameter sets in the table on the left were used for exploring the general OMP model behavior with both, correlated and independent spikes, yielding $n = 2 \times 8640 = 17280$ runs. Simulations with $\sigma_D = 10$ ms were used to create *Figure 3A*, while other runs from this set are used in *Figure 3—figure supplement 1* and *Figure 4—figure supplement 2*. The parameter sets on the right ($n = 3200$) were used in *Figure 3E*, in which we comprehensively examined the two most important parameters, $\tau_G$ and $\tau_s$.

| Parameter | Values | Parameter | Values |
|---|---|---|---|
| $\lambda_M$ | 0.01, 0.02, 0.05, 0.1, 0.2 | $\lambda_M$ | 0.02, 0.05 |
| $\lambda_A$ | 0.01, 0.1, 1 | $\lambda_A$ | 0.01 |
| $\lambda_H$ | $10^{-5}$, $10^{-6}$ | $\tau_G$ | 4, 6, 8, …, 82 |
| $N_A$ | 10, 25 | $\sigma_D$ | 10 |
| $N_O$ | 5, 10 | $\tau_S$ | 5, 10, 15, … 195, 200 |
| $\tau_s$ | 25, 50, 100, 200 | $\sigma_j$ | 1 |
| $n_r$ | 5 | $n_r$ | 10 |
| | | $T_e$ | 60 |

**Appendix 1—table 2.** Parameter sets used in our OMP model simulations.
On the left are parameter sets used to create *Figure 3B, C and D* exploring the influence of the number of OL, $N_O$, in the OC ($n = 576$ runs). On the right are parameter sets exploring mixed signals shown in *Figure 4A, B and C*, which were repeated in three scenarios: (1) two equally sized groups of axons, one carrying correlated and the other independent spikes, (2) two equal and separately correlated groups, and (3) one independent and four equal separately correlated groups. Scenario 1 was used in *Figure 4A and B*, and scenarios 2 and 3 are shown in *Figure 4C*. Two additional matched sets of runs are performed for "pure" correlated or independent spikes, matched in terms of $N_A$ within a group, yielding a total of $n = 6912 + 2 \times 2304 = 11520$ runs.

| Parameter | Values | Parameter | Values |
|---|---|---|---|
| $\lambda_A$ | 0.01, 0.1, 1 | $N_A$ | 20, 50 |
| $N_O$ | 1, 2, 5, 10 | $N_O$ | 5, 10 |
| $\sigma_D$ | 5 | $\tau_s$ | 25, 50, 100, 200 |
| $t_R$ | 0, 30 | $n_r$ | 5 |
| $\tau_s$ | 25, 50, 100, 200 | | |
| $n_e$ | $500/N_O$ | | |

**Appendix 1—table 3.** Parameter sets used in our OMP model simulations.

On the left are parameter sets used to explore the influence of homeostatic rate ($n = 2520$) shown in *Figure 4D and E* and *Figure 4—figure supplement 2*. Parameters used in *Figure 4—figure supplement 2E, F*, exploring the influence of the spiking rate variability on synchronization ($n = 1296$).

| Parameter | Values | Parameter | Values |
|---|---|---|---|
| $\lambda_M$ | 0.01, 0.02, 0.05, 0.1, 0.2 | $\lambda_M$ | 0.02, 0.05, 0.1 |
| $\lambda_H$ | $10^{-2}$, $10^{-3}$, ... , $10^{-7}$, $10^{-8}$ | $\lambda_A$ | 0.1 |
| $N_A$ | 10, 20 | $N_A$ | 10, 20 |
| $\sigma_D$ | 10 | $\tau_G$ | 10, 20, 30, 50 |
| $\sigma_j$ | 1, 3 | $\sigma_D$ | 5, 3, 7 |
| $n_r$ | 5 | $\sigma_j$ | 1 |
| | | $\sigma_s$ | 0, 1, 2, 5, 10, 20 |
| | | $T_e$ | 20 |

**Appendix 1—table 4.** Two parameter sets for simulations used in *Figure 1—figure supplement 4A* (left) and *Figure 1—figure supplement 4B* (right), comparing theoretical predictions for the pure Poisson synchronized spikes with $\sigma_j = 0$ to the values obtained in simulations with non-zero $\sigma_j$. For the table on the left, with $t_R = 0$ ms there were $n = 2916$ runs and for the one on the right, with refractory Poisson spikes ($t_R = 30$ and $80$ ms) there were $n = 288$ runs.

| Parameter | Values | Parameter | Values |
|---|---|---|---|
| $\lambda_M$ | 0.01, 0.1 | $\lambda_M$ | 0.01, 0.1 |
| $\lambda_A$ | 0.1, 0.01, 0.001 | $\lambda_H$ | $10^{-7}$ |
| $\lambda_H$ | $10^{-7}$ | $\tau_G$ | 5, 10, 30 |
| $N_A$ | 2, 5, 10 | $\sigma_D$ | 5 |
| $N_O$ | 1, 5, 10 | $t_R$ | 30, 80 |
| $\tau_G$ | 5, 10, 30 | $\tau_s$ | 100, 200 |
| $\sigma_j$ | 0.1, 1, 3, 5 | $\sigma_j$ | 0.1, 1, 3, 5 |
| $n_r$ | 3 | $n_r$ | 2 |
| $n_e$ | 1000 | $n_e$ | 1000 |
| $T_e$ | 20 | $T_e$ | 5 |

## Appendix 2

### Predictions for two axons with regular spiking

In the case of regular spiking, where the ISI times are distributed as $p_{\text{ISI}}(t) = \sum_k \delta(t - \tau_s)$, *Equation 1* reduces to evaluating the sum,

$$G_a(\delta t_a) = \sum_{k=0}^{\infty} R(k\tau_s + \delta t_a). \tag{23}$$

The sum in *Equation 23* can be easily evaluated if we note that for our choice of $R(t)$ (*Equation 7*), the response function can be written as the difference of two exponential functions, (for clarity, abbreviating $\tau_G$ as just $\tau$).

$$R(t) = e^{-\frac{t}{\tau}} - e^{-\frac{2t}{\tau}}, \tag{24}$$

which reduces to evaluating two geometric sums, yielding,

$$G_A(\delta t | \tau, \tau_s) = \frac{2e^{\frac{\tau_s - 2\delta t}{\tau}} \left( e^{\delta t/\tau} + e^{\frac{\delta t + \tau_s}{\tau}} - e^{\tau_s/\tau} \right)}{\tau \left( e^{\frac{2\tau_s}{\tau}} - 1 \right)}, \tag{25}$$

where $\delta t$ is the time to the most recent spike, and $G_a$ is replaced with $G_A$, indicating that the form of $G_A$ will be the same for different axons, but $\delta t$ on separate axons will differ. In other words, the contributions to the global $G(t)$ from different axons is only going to differ through the difference of their times to the most recent spikes, $\delta t_a$, i.e., $G_a(\delta t_a) \equiv G_A(\delta t_a)$. Assuming $t = 0$ coincides with one of the spikes on a given axon we can write, $\delta t \equiv t \pmod{\tau_s}$. When combining $G_A(\delta t_a)$ signals from all axons, generally only one of them can be chosen as such reference, while $\delta t$ for others will be expressed in terms of fixed delays between them, and will require considering separately different orderings, depending on the fixed temporal delays between them. For example, in the simple case, $N_A = 2$, for a given fixed delay, $t_d$, between two regular spiking trains (*Figure 1—figure supplement 3A*), we need to distinguish between the cases $\delta t \leq t_b$ versus $\delta t > t_b$, where $t_b = \tau_s - t_d$. In *Figure 1—figure supplement 3A and B*, we color the "leading" axon (the one whose spikes arrive first) as "green", while the other one, referred to as the "lagging" axon, is colored red. We can see that the time to the last spike will depend on the magnitude of $\delta t_1$. When $\delta t_1 \leq t_b$,

$$G(t) = G_A(\delta t_1 | \tau_G, \tau_s) + G_A(\delta t_1 + t_d | \tau_G, \tau_s)$$

and for $\delta t > t_b$,

$$G(t) = G_A(\delta t_1 | \tau_G, \tau_s) + G_A(\delta t_1 + t_d - \tau_s | \tau_G, \tau_s).$$

Since $t = 0$ is set for "red" (lagging) spikes, they will occur at times, $t = k\tau_s$, while "green" spikes will occur at $t = k\tau_s + t_d$, and, according to the OMP model, the amount of $M$ will be proportional to $G(t)$ at those times. This is plotted in *Figure 1—figure supplement 3B*, and we see that the lagging axon will higher concentration of myelin-promoting factor $M$ than the "green" (leading), which will synchronize them. This, however, switches when $\tau_s < 2t_d$, i.e., for critical spiking frequency, $f_s^{\text{crit}} = 1/(2t_d)$, which in this case $f_s^{\text{crit}} = 50$ Hz. Such "reversed myelination" can also be seen in *Figure 1—figure supplement 3B*, indicated by "orange" and "red" regions. The dashed lines indicate the contours where the ratios are equal to 0.25, 0.75, and 1, as labeled. The $r_2 = 1$ contour occurs for $f_s^{\text{crit}}$. Evaluating $M_a$ for $N_A > 2$ and in the presence of jitter requires more elaborate calculations, that are left to be addressed outside of the current manuscript.

