## [Editor Report]

This paper presents a new mathematical model describing biologically plausible feedback that glial cells might use to properly modify the conduction velocity in axons and promote optimal timing of neural impulses through changes in myelination. This work provides an important step forward by providing the theory for myelin-mediated neuronal plasticity.

---

## [Decision Letter]

**Decision letter after peer review:**

Thank you for submitting your article "Oligodendrocyte-mediated myelin plasticity and its role in neural synchronization" for consideration by *eLife*. Your article has been reviewed by 2 peer reviewers, and the evaluation has been overseen by a Reviewing Editor and Timothy Behrens as the Senior Editor. The following individual involved in the review of your submission has agreed to reveal their identity: Lee Susman (Reviewer #2).

Essential revisions:

(1) OLs are electrically non-excitable cells, have leak K^+^ currents, usually sit at the resting membrane potential of -90 mV, and are very difficult to depolarize. What experimental evidence is that a single OL can integrate signals from several axonal segments it myelinates? Also, after integrating these signals, what type of feedback can an OL provide to an axon to adjust axonal conduction velocities (CV)?

(2) The model proposed by the authors assumes that neuronal spikes induce the release of a potent signaling factor G, which needs to be faster than 80 ms. But, OLs have not been shown to generate such fast electrical signals or even ca^2+^ spikes. Also, OLs and even the progenitors of OLs (OPCs) don't seem to integrate the local ca^2+^ signals in their processes in a cell-wide manner.

(3) Another assumption is that an OL can integrate G at various segments and, in turn, lead to a local release of an unknown myelin promoting factor, M. M is suggested to be fast acting as well and can promote differential myelination by balancing myelin addition and a steady-state myelin removal, and can modulate axonal CV. In the OMP model, the primary concern is the hypothetical assumption of the existence of the factors G and M. The authors don't suggest or relate to any past experimental work/s where the readers can get any clue on the identity or possible existence of such factors in a real-world biological setting.

(4) The authors highlight that an OL can myelinate up to 50 axons; mostly, these axons are unique, and the same axon is not myelinated twice. The selection of these 50 axons is seemingly random, as there is no experimental evidence that an OL follows a specific rule to myelinate axons (even more so during development). In addition, the axons myelinated by an OL can be from excitatory and inhibitory neurons, and these neurons have very different spiking properties. Thus, to increase CV at a particular axonal segment, OL will need to communicate and coordinate with neighboring OLs, and it is unclear how that happens.

(5) Recent studies on rodents convincingly show that myelin sheath extension and retraction occur. But this constitutes a small part of the adaptive myelination, a more common form of new myelin addition is the generation of new OLs through the differentiation of OPCs. In addition, astrocytes (much closer to synapses) might sense neuronal activity and release molecule A (e.g., a gliotransmitter), which can promote OPC differentiation and/or formation of new myelin internodes by the existing OLs. The authors touched on an exciting topic related to fine-tuning myelination by OLs in a neuronal activity-dependent manner. Still, the model's assumptions are too far from the actual biology. Although it is essential to have simplified assumptions for a complex mathematical model, a model without a robust prior underpinning of experimental evidence tends not to push forward the in-silico research.

(6) One of the concrete predictions from the model is that optimal synchronization occurs in the regime of low firing rates. How does this result depend on the model choice? For example, does this change in the more general model OMP-n? Does it depend on the homeostatically set myelin removal rate? Do the authors have an idea as to the mechanism underlying this optimal point?

(7) The mathematical model presented in this work has several interacting compartments, even in the simple case of OMP-1, and the reader can easily get confused about the overall architecture. It might be beneficial to put more emphasis in Figure 1 on the schematic presentation of the basic model components (G, R, M, λ, and tau). If the OMP-n model is not discussed at length, its graphical presentation may be put elsewhere, perhaps in an appendix.

---

## [Author Response]

Essential revisions:(1) OLs are electrically non-excitable cells, have leak K^+^ currents, usually sit at the resting membrane potential of -90 mV, and are very difficult to depolarize. What experimental evidence is that a single OL can integrate signals from several axonal segments it myelinates? Also, after integrating these signals, what type of feedback can an OL provide to an axon to adjust axonal conduction velocities (CV)?

This helpful and important query asks several different questions that we address separately below ((a)-(c)). We have also modified the manuscript to included most of the descriptions provided here and all the references mentioned below:

(a) Regarding oligodendrocytes as non-excitable cells: These comments conform to the traditional view, but this has been revised based on evidence obtained in the last decade. OLs can undergo depolarizations and can also behave as excitable cells (in the sense of producing sudden transitions akin to action potentials), although the latter property is not essential for our OMP-1 model. We agree that some of these issues we need to discuss more effectively, and we do so in the new version of the manuscript, mainly in the S and I section. Oligodendroglia are responsive to action potentials through several mechanisms, including neurotransmitters and depolarization by extracellular potassium. Oligodendrocyte progenitor cells (OPCs) are frequently coupled synaptically to axons via excitatory, glutamatergic (Kukley et al., 2007; Ziskin et al., 2007) and inhibitory GABAergic connections (Maldonado and Angulo, 2015). This rapid signaling enables the glial cells to respond to different patterns of action potentials with different functional effects on cell differentiation and proliferation (Nagy et al., 2017). Despite their low transmembrane resistance, as correctly noted by the reviewer, oligodendrocytes that have formed compact myelin on axons can be depolarized by axonal action potential firing. Additionally, we already cite an important finding in the original manuscript (which is not used in our OMP-1 model) that theta burst stimulation of axons depolarizes oligodendrocytes forming mature myelin in the CA1 region of hippocampus via glutamate signaling and potassium channels (Yamazaki et al., 2007). Direct depolarization of these oligodendrocytes via patch clamp electrode increases conduction velocity in myelinated CA1 axons within seconds of stimulation showing an activity-dependent signaling from the axon to the myelin sheath that has functional consequences for action potential propagation velocity. In addition to potassium signaling and membrane depolarization, several types of signaling mechanisms between compact myelin and the underlying axon have been identified that enable activity-dependent interactions (Stys, 2011).

(b) Regarding experimental evidence that oligodendroglia can integrate several axons it myelinates: this has been shown by calcium imaging in several different studies, and we have revised the text to clarify this point: Confocal imaging of local intracellular calcium responses in myelinating oligodendrocytes in cell culture together with imaging local translation of myelin basic protein, show that action potentials in axons cause local calcium transients in oligodendroglial via vesicular release of glutamate from axons acting on NMDA and mGluR glutamate receptors on the oligodendrocyte cell process (Wake et al., 2011). This promotes the formation of an axoglial signaling complex, triggering the local translation of myelin basic protein to initiate myelination (Wake et al., 2011), preferentially on the electrically active axons (Wake et al., 2015). This local signaling can trigger myelin synthesis rapidly, within minutes (Wake et al., 2011). It also demonstrates integration of the calcium signals, e.g. in Wake et al., 2011 Figure S7, which shows that:

“Simultaneous local calcium responses in individual oligodendroglial cell processes are observed more frequently than expected by probability, implying cooperativity of calcium transient generation between processes or propagation of transients from one process into another. When somatic intracellular calcium is high, there is a larger probability for one or more cell processes to have an elevated intracellular calcium response. The probability of somatic calcium elevation increases when more processes have high intracellular calcium transients simultaneously, indicating that simultaneous activity in several cell processes of the same oligodendrocyte triggers calcium elevations in the soma (a global response), possibly influencing gene expression.”

The latency of local calcium signaling in oligodendrocytes is slower than what is required by our model (but, faster than 500 ms), but this reflects the slow kinetics of GCaMP2 calcium indicator. The actual signaling kinetics is much faster; the signaling from axons to oligodendrocytes in this case is not via synaptic vesicles, but rather by action potential induced exocytosis of glutamatergic vesicles at axon varicosities (Wake et al., 2015). Other activity-dependent signaling molecules released from axons firing action potentials, notably ATP and adenosine, activate purinergic receptors that are expressed throughout the oligodendrocyte cell membrane to cause global increases in cell calcium that activate gene expression to promote myelination (Stevens et al., 2002).

in vivo imaging in zebrafish show similar responses (Hines et al., 2015; Mensch et al., 2015), and further indicate that myelin sheath elongation during development is regulated by the kinetics of calcium transients in oligodendrocytes that are evoked by neuronal activity (Krasnow et al., 2018). Myelin sheath elongation is promoted by high frequency calcium transients, and sheath shortening is associated with low frequency calcium transients. The same study shows that local calcium transients in oligodendrocyte cell processes are independent from one another and that the local calcium signals can be integrated within the cell to trigger global calcium responses in the cell body via temporal summation (Krasnow et al., Supp Figure 1).

(c) Regarding the type of activity-dependent feedback to the axon from oligodendrocytes that can affect conduction velocity, there are numerous mechanisms. We now cite a recent review which reviews these interactions (Munyeshyaka and Fields, 2022), however this communication is not something that our model needs or relies on. The OLs are the cells that myelinate the axons directly and consequently modify their conduction velocity. They do not necessarily need to provide feedback to axons to change CV, although such communication can also have effects on the CV (redistribution of channels, etc). The main biological assumption in our model is that the neural activity in the axons is sensed and integrated by the OLs and based on this information myelin can be differentially distributed across different axons. In our mathematical model we propose a minimal model of interactions that could lead to adaptive synchronization, which needs one global factor, G, and the local myelin promoting factor, M; by local we mean that its concentration will differ in different OL processes. At this stage their identity is only a speculation, but as stated, we now provide additional references that support our belief that this model is biologically plausible.

Thank you for these questions and for the opportunity to include this important information in the edited manuscript.

(2) The model proposed by the authors assumes that neuronal spikes induce the release of a potent signaling factor G, which needs to be faster than 80 ms. But, OLs have not been shown to generate such fast electrical signals or even ca^2+^ spikes. Also, OLs and even the progenitors of OLs (OPCs) don't seem to integrate the local ca^2+^ signals in their processes in a cell-wide manner.

We disagree with some of the statements in #2 and now provide references showing evidence that the transient responses to neural activity in OL do occur, including Ca^++^ response, as elaborated in the response #1b. As mentioned there, the observed characteristic time for these events is indeed slower than what would be required for optimal performance of the OMP model (< 80 ms versus 500 ms that is observed) but part of this is the slow characteristic time of calcium indicators. In (Wake et al., Science 2011) was shown that the action potentials propagating along axons induce local and global calcium responses in oligodendrocytes forming myelin on unmyelinated axons in OLs via vesicular glutamate release and also via non-vesicular release of ATP from axons (Wake et al., Nat. Comm. 2015; Fields and Ni Sci. Signaling 2010).

Different neurotransmitters and neurotransmitter release mechanisms can produce either local or global calcium responses in oligodendrocytes in response to axonal firing. At the same time, oligodendrocytes can integrate local calcium signals in the oligodendrocyte cell processes to produce a global calcium response. Local calcium transients induced by action potential firing in oligodendrocyte cell processes are blocked by botulinum toxin, but a global somatic calcium response in the cell body persists (Wake et al., 2011) due to purinergic receptors that are expressed throughout the oligodendroglial cell membrane. Global calcium responses in the oligodendrocyte can regulate gene expression whereas local calcium signals can control myelination and other cell processes locally and selectively in different oligodendrocyte cell processes.

(3) Another assumption is that an OL can integrate G at various segments and, in turn, lead to a local release of an unknown myelin promoting factor, M. M is suggested to be fast acting as well and can promote differential myelination by balancing myelin addition and a steady-state myelin removal, and can modulate axonal CV. In the OMP model, the primary concern is the hypothetical assumption of the existence of the factors G and M. The authors don't suggest or relate to any past experimental work/s where the readers can get any clue on the identity or possible existence of such factors in a real-world biological setting.

In our model we do not suggest that *M* needs to be fast acting. In fact, we show that the characteristic rate for conversion of *M* into myelin does not influence our results much (see Figure 4 - figure supplement 2D), and hence, a simplification of the algorithm to infinitely fast myelination is introduced, mainly as an example of a leaner and computationally less demanding way of implementing and studying OMP but is not something that OMP model needs to function. In fact, this faster implementation has drawbacks of being numerically unstable in some situations, as emphasized in the text. In this paper we wanted to focus on creating a model that conceptually produces the desired results, i.e., synchronize inputs to a given target, but also retains some biological realism. We again emphasize that by biological realism we mean that the model can be realized in principle, and only loosely based on experimental evidence, as described above. Other models of myelin plasticity presume that the feedback on arrival times at target are available to oligodendrocytes, far away from it, which cannot be justified even in principle.

(4) The authors highlight that an OL can myelinate up to 50 axons; mostly, these axons are unique, and the same axon is not myelinated twice. The selection of these 50 axons is seemingly random, as there is no experimental evidence that an OL follows a specific rule to myelinate axons (even more so during development). In addition, the axons myelinated by an OL can be from excitatory and inhibitory neurons, and these neurons have very different spiking properties. Thus, to increase CV at a particular axonal segment, OL will need to communicate and coordinate with neighboring OLs, and it is unclear how that happens.

In this manuscript the OL chain is not a detailed simulation of individual OLs and their detailed connectivity, but each effective “OL” in the chain represents the population of the OL cells that on a given segment of axonal bundle pathway myelinates a given set of axons and presumed to react in some fashion to the spike patterns at that location. We now expand on this in the section titled “OL Chain”:

“As already emphasized, the OC depicted in Figure 2 does not imply literally that there are N_O_ OL cells along the axons, but rather that there are N_O_ segments, representing N_O_ different populations of oligodendrocyte cells myelinating different portions of the axonal bundle, which modulate the delays locally. Assuming that all cells within the same segment will receive the same pattern of spikes and respond to it in the same way, they all can be governed by a single OMP equation. Individual oligodendrocyte cells, in fact, would not be able to modify the delays effectively and independently from other oligodendrocyte cells in the same location, as they would not be able to form tight nodes of Ranvier, considering that OLs prefer not to myelinate the same axon multiple times. Neighboring OL cells are then needed to stack their processes, reducing the width of the nodes of Ranvier and in this way greatly increasing the CV, i.e., reducing the conduction delays.”

In the current manuscript we do not explore the effects of random connectivity and presume that the OL cells that myelinate only a small fraction of the selected axons won’t have detrimental effect on synchronization. In general , the OMP model does not presume that all actual/biological OLs have to myelinate precisely the same axons everywhere along a given axon bundle, although we do speculate in the discussion that the innervation of similar subset of axons might be happening by having the processes on the synchronous axons less likely to be withdrawn or more likely to be formed; but the ability of OMP model to synchronize inputs does not rely on that assumption. In our simulations we simulate OL chains with up to 10 OLs in a sequence, which is vastly lower than the number of OLs that would actually participate in any given path. We took the shortcut in our current exploration, aiming to show that OMP model can work in principle, and presumed that we only need to simulate the subset of OLs that have the same/similar pattern of myelination, which would be the most efficient way to synchronize given spike trains. OLs that myelinate mostly other axons or axons that do not have correlated signals would not have large influence in modifying the conduction times. To precisely address the effect of random connectivity we need to simulate thousands of OLs using random or some other more realistic connectivity for each segment of OL chain, which would be orders of magnitude more demanding computational task. Because in the current manuscript we explored tens of thousands different settings of parameters, we could not afford conducting such large-scale simulations and took the above-mentioned shortcut. Our idealized OL chain does show that within any subset of axons an OL innervates, the OL will preferentially synchronize those axons carrying time-locked activity. We now emphasize these issues in the Discussion (second paragraph) and in the Materials and methods sub-section on “OL-axon Connectivity”.

Regarding the question of axons arising from excitatory and inhibitory neurons, which have very different frequencies, is a valid objection to our model, and is addressed in Figure 4—figure supplement 2E and F, where we show that the OMP model does not work well when axons with significantly different spiking rates are involved. Our model assumes OL myelinating excitatory and inhibitory axons separately. When sources have significantly different frequencies then it is also hard to envision what the spike synchrony would mean. The question whether OLs that myelinate both slow spiking and fast spiking axons can synchronize each separately is left for future explorations.

Evidence also shows that oligodendrocytes do not myelinate axons indiscriminately; rather many axo-glial signaling mechanism determine which axons are myelinated and when. Axon caliber and the electrical activity in axons are notable examples. Current evidence is that oligodendroglia are a very heterogeneous group, with different properties that enable them to selectively respond to different types of neurotransmitters and other signals from neurons to have specific interactions with neurons of different types, stages, and physiological state; this includes both myelin plasticity and other types of activity-dependent plasticity. While inhibitory axons can also be myelinated, it is not clear that the same individual oligodendrocyte myelinates both excitatory and inhibitory axons within its domain.

(5) Recent studies on rodents convincingly show that myelin sheath extension and retraction occur. But this constitutes a small part of the adaptive myelination, a more common form of new myelin addition is the generation of new OLs through the differentiation of OPCs. In addition, astrocytes (much closer to synapses) might sense neuronal activity and release molecule A (e.g., a gliotransmitter), which can promote OPC differentiation and/or formation of new myelin internodes by the existing OLs. The authors touched on an exciting topic related to fine-tuning myelination by OLs in a neuronal activity-dependent manner. Still, the model's assumptions are too far from the actual biology. Although it is essential to have simplified assumptions for a complex mathematical model, a model without a robust prior underpinning of experimental evidence tends not to push forward the in-silico research.

We disagree that our results are far from actual biology. While OPC could be a prevalent mechanism in modifying myelin content it lacks the ability to gain feedback and adjust CVs differentially between axons in appropriate manner, i.e., a manner that leads to synchronization at the target. However, we also emphasize now that OPC differentiation can be an important mechanism in the formation of the OL chain, by increasing the number of OLs (see the last paragraph in the S and I section). There we cited reference to our early studies that were among the first to demonstrate that proliferation and differentiation of oligodendroglia are regulated by action potential firing in axons via adenosine signaling (Stevens et al., Neuron 2002), ATP and LIF (Ishibashi et al., Neuron 2006).

(6) One of the concrete predictions from the model is that optimal synchronization occurs in the regime of low firing rates. How does this result depend on the model choice? For example, does this change in the more general model OMP-n? Does it depend on the homeostatically set myelin removal rate? Do the authors have an idea as to the mechanism underlying this optimal point?

In this work we only studied the OMP-1 and have now removed OMP-n model from the main figure, as suggested in comment #7. Since OMP-n requires longer cascades of G-factors and events we envision that it would also work better at low firing rates; however, this will largely depend on the details of the OMP-n model in which other triggering events can be introduced. Due to the potential complexity, the OMP-n model is not studied in this work. We do not address the biological mechanisms underlying the homeostatic balance and we emphasize this in the Discussion. Although we assume that this balance occurs at the level of a single OL (or rather, a small OL population at a given axonal segment), it is natural to assume that at some level the homeostatic mechanism is needed to balance the removal and the addition of myelin in a healthy nervous system.

(7) The mathematical model presented in this work has several interacting compartments, even in the simple case of OMP-1, and the reader can easily get confused about the overall architecture. It might be beneficial to put more emphasis in Figure 1 on the schematic presentation of the basic model components (G, R, M, λ, and tau). If the OMP-n model is not discussed at length, its graphical presentation may be put elsewhere, perhaps in an appendix.

We agree and now have modified the figure to include only the OMP-1 model, and the depiction of the OMP-n model is moved to a supplemental figure. This also allowed us to refer to OMP-1 model simply as the OMP model (within this manuscript).